

**A simple weather generator for applications with limited data availability: TEmpotRain 1.0 for temperatures, extraterrestrial radiation, and potential evapotranspiration**

Gerrit Huibert de Rooij

Helmholtz Centre for Environmental Research – UFZ, Soil System Science Dept.,

Theodor−Lieser−Strasse 4, 06120 Halle (Saale), Germany

E−mail: gerrit.derooij@ufz.de

**Abstract**

A weather generator is introduced that has a Bartlett−Lewis rainfall generator in which storms with exponentially distributed time intervals between their starting times consist of cells of which the intervals between their starting times are exponentially distributed, and their durations and rainfall rates are both gamma−distributed. Each day is either overcast or clear, with the probability of a cloudy day depending on the daily rainfall. A temperature generator uses a sinusoidal annual signal of which the mean and the

amplitude are both normally distributed. For overcast days, the amplitude is reduced. Superimposed on this signal is a first−order autoregressive model with independently identically normally distributed shocks for the daily mean temperature, which is assumed to be the average of the daily minimum and maximum temperature. The difference between the daily mean and extremes follows a lognormal distribution, the standard deviation of which is reduced for overcast days. The daily extraterrestrial

radiation, mean and extreme temperatures, and, for one of the two models used, the 30−day rainfall sum, determine the daily potential evapotranspiration. To permit the generation of very long time series, leap years are taken into account. One hundred years of weather data were generated for two contrasting climates. The results show that the choice of the evapotranspiration model is consequential for temperate climates. Additional calculations demonstrate the effect of the daily temperature fluctuations on the

potential evapotranspiration. Standard computational resources (laptop) suffice to run the weather generator. The Fortran90 source codes, input file formats, and user manual are provided.



## 1. Introduction

Models for soil water flow, solute transport, and plant growth (e.g., Kroes et al., 2000; Šimůnek et

al., 2016) are used to optimize irrigation (e.g., Ma et al., 2011), estimate groundwater recharge (e.g.,

Scanlon et al., 2003), etc. They require meteorological input data, most notably precipitation, air

temperature, and potential evapotranspiration. Poorer and/or less developed regions of the world often

lack the monitoring and research infrastructures for long−term data collection required for the

establishment of useful meteorological time series. Paucity of up−to−date time series is exacerbated by

climate change as historic data records become decreasingly representative for future conditions,

diminishing their usefulness for scenario studies even in regions with ample weather monitoring stations.

Unfortunately, both factors (lack of data and climate change impact) can operate in tandem. As a case in

point, the regions that face the largest risk of freshwater shortage are predominantly located in North

Africa, the Arabian Peninsula, and the region from the eastern Mediterranean coast to India (FAO 2016,

Fig. 1). Both the lack of data and their limited value in a changing climate compromise our ability to

model soil water flow and crop development under local conditions. This in turn makes it more difficult to

optimize agricultural water use (e.g., Howden et al., 2007; Chartzoulakis and Bertaki, 2015), estimate

groundwater recharge (e.g., Toews and Allen, 2009), or adapt water management strategies to the

changing climate (e.g., Shresta et al., 2015).

In view of this, it will increasingly be necessary to generate artificial weather records based on

limited data combined with expert judgement of local conditions and trends, or on future climate

projections (e.g., Kirtman et al., 2013), either because local data are not available or the target time period

for scenario studies requires future climatic conditions to be considered. Sophisticated stochastic weather

generators (e.g., Breinl et al., 2015) are increasingly powerful, but difficult to operate with scant data. The

web−based MarkSim DSSAT weather generator (see links in Jones and Thornton, 2013) uses a suite of 17

General Circulation Models to generate daily values of various weather parameters at any desired location

and future year, relying on four scenarios developed by the Intergovernmental Panel on Climate Change.



For each year, it can generate up to 99 realizations. The underlying weather generator relies for values of

its 117 parameters on regression equations for 702 classes of world weather (Jones and Thornton, 2013).

The model therefore does not require the large volume of data needed to calibrate advanced stochastic

weather generators but can only generate weather for climates within the envelope of current climates.

A relatively simple, stand−alone weather generator with parameters with well−defined physical

meaning has the advantage that the generated weather can be 'constructed' by setting the parameters to

values that represent the current or expected future weather conditions without the need for calibration on

inadequate or non−existing data sets. Another advantage is that such a model can explore climates outside

the range of currently prevailing conditions anywhere on the planet. Its stand−alone nature allows

applications in locations with limited resources and infrastructure. The objective of this paper is to present

a weather generator consisting of a suite of connected models for generating daily values of precipitation,

air temperature, extraterrestrial radiation, and potential evapotranspiration for essentially unlimited

periods of time. Its output matches the input data required by a popular model for water flow and solute

transport in vegetated soils.

## 2. The weather generator TEmpoTrain: underlying models

### 2.1. Rainfall

Artificial rainfall records are generated using a modification of the Bartlett−Lewis model

originally proposed by Rodriguez−Iturbe et al. (1988). The intervals between starting times of rain storms

have an exponential distribution. Within each storm, rain cells of variable duration and with different but

constant rainfall rates occur. The total rainfall rate at any given time equals the sum of the rainfall rates of

all rain cells that are active at that time. The model has a gamma−distributed parameter $\eta$ that defines the

exponential distribution of the cell duration in a storm. This distribution has shape parameter $\alpha$ and rate

parameter $\nu$ (d) (Table 1). The mean of $\eta$ therefore equals $\alpha / \nu$. Indirectly, $\eta$ also governs the duration of

storms and the intervals between the times at which rains cells within a storm start. The rainfall rate in the




cells is also gamma−distributed (Onof and Wheater, 1994), with shape parameter $p$, rate parameter $\delta$ (d mm$^{-1}$) (Table 1), and mean $p / \delta$.

Because the Gamma distribution can occasionally produce unrealistically long−lasting rain cells, a truncated version is used that rejects values of $\eta$ below a threshold $\varepsilon$ (Onof et al., 2013). By setting $\varepsilon$ to zero, the untruncated version of the model is obtained. Pham et al. (2013) give full details of these untruncated and truncated modified Bartlett−Lewis models with gamma−distributed rainfall rates (MBLG and TBLG model, respectively, in their terminology). The model requires 8 parameters for each of the

user−prescribed periods into which a year can be divided (Table 1). The time unit used is day; the length unit is arbitrary. For clarity and consistency with the model for evapotranspiration, mm is used as a unit in the paper.

        Untruncated and truncated versions of the rainfall generator with exponentially distributed rainfall rates of the rain cells are also provided for completeness. In that case, parameters $p$ and $\delta$ in Table 1 are

replaced by a single parameter $m$ (mm d$^{-1}$), the reciprocal of the mean rainfall rate of the cells. The exponential distribution of rainfall rates does not reproduce extreme rainfall very well (Pham et al., 2013), so the models with $m$ will not be used in this study.

### 2.1. Temperature

The average daily temperature is assumed to have a sinusoidal trend over the year. To allow interannual variation, the year−to−year fluctuations of the annual mean temperature as well as the amplitude of the sinusoidal signal are independently normally distributed. The de−trended daily mean temperatures are first−order autoregressive (AR1; Tsay, 2010, p. 37−40) with zero mean and independent, normally distributed shocks. This leads to the expressions for the mean daily temperature:

$\bar{T}_{i,\text{trend}} = N_1(\bar{T}_a, \sigma_T) + \frac{A_{n,i}}{A_c} N_2(A_c, \sigma_a) \left\{ \sin\left[\frac{2\pi(\psi + i \bmod 365 - 0.5)}{365}\right] \right\}$           (1a)

$\bar{T}_i = \bar{T}_{i,\text{trend}} + \phi\left(\bar{T}_{i-1} - \bar{T}_{i-1,\text{trend}}\right) + N_3(0, \sigma_m), \ 0 \leq \phi < 1$           (1b)



where $\bar{T}_i$ (°C) is the mean temperature of day $i$, $\bar{T}_{i,\text{trend}}$ (°C) is the daily mean temperature predicted purely

from the annual fluctuation, $\bar{T}_a$ (°C) is the mean annual temperature, $\sigma_T$ (°C) is the standard deviation of

the year−to−year variation of $\bar{T}_a$, $A_{n,i}$ (°C) is the amplitude of the annual temperature that can take the

value $A_c$ on clear days and the value $A_o$ on overcast days, $\psi$ (d) is the temporal shift of the annual

temperature cycle, $\phi$ determines the persistence of the dynamic dependence of the time series, $\sigma_a$ (°C) is

the standard deviation of the variation of the annual amplitude for clear days, $\sigma_m$ (°C) is the standard

deviation of the shocks, and $N_j(\mu,\sigma)$ is a normally distributed variate with mean $\mu$ and standard deviation

$\sigma$. Realizations of random variates $N_1$ and $N_2$ are drawn once for every year, while a value for $N_3$ is

redrawn every day.

The range between the daily minimum and maximum temperatures is also assumed to be random

and identically independently distributed, but with a lognormal distribution. The minimum and maximum

daily temperatures are generated by superimposing lognormally distributed fluctuations on $\bar{T}_i$:

$$T_{\substack{\min,i \\ \max,i}} = \bar{T}_i \mp e^{N_4(\mu_f,\sigma_{f,n})} \tag{2}$$

where $T_{\min,i}$ (°C) is the minimum and $T_{\max,i}$ (°C) the maximum temperature of day $i$, and $\mu_f$ is the mean and

$\sigma_{f,n}$ the standard deviation of the natural logarithm of the probability distribution of the daily fluctuations.

A value for $N_4$ is drawn every day.

Other models for daily extreme temperatures found that low−order autoregressive moving average

(ARMA) models work well (e.g., Kalvová and Nemešová, 1998). These models analyzed the statistics of

air temperature without considering other weather variables. The model presented here reduces the

temperature fluctuations for cloudy days, and links the probability of cloud cover to the amount of

precipitation. This implicitly creates a degree of autocorrelation in the temperature record, in addition to

that explicitly introduced through parameter $\phi$ in Eq. (1b).

Cloud cover is expected to lead to lower summer temperatures and higher winter temperatures

(the range between which is governed by $A_{n,i}$ in Eq. (1a)), compared to clear days. It is also assumed to



reduce the daily temperature range, governed by $\sigma_{f,n}$ in Eq. (2). The temperature model can therefore

accept one pair of values ($A_c$, $\sigma_{f,c}$) for clear days and another pair ($A_o$, $\sigma_{f,o}$) for overcast days, with the

values for the latter smaller than for the former. The probability of a day being overcast is likely to be

related to the amount of rain during that day. Small amounts of rainfall not necessarily imply complete

cloud cover during the day. Intermediate amounts of rainfall are often associated with the passage of a

weather front and thus carry a high likelihood of complete cloud cover all day long. Heavy rainfall can

either be caused by convective showers, and thus do not preclude a largely clear day, or be associated with

heavy storms or gales with complete cloud cover. The continuum between zero and partial cloud cover

during varying parts of the day and their relation to daily rainfall sums is simplified to a three−level

staircase function with five parameters in total:

$$f_i(\text{overcast}) = \begin{cases} f_1, & P_i \leq P_l \\ f_2, & P_l < P_i \leq P_h \\ f_3, & P_i > P_h \end{cases} \tag{3}$$

where $f_i$(overcast) is the probability that day $i$ is cloudy, $f_1$, $f_2$, and $f_3 \in [0,1]$ are parameters that

determine this probability for the specified conditions, $P_i$ (mm) is the total amount of rainfall on day $i$, and

$P_l$ and $P_h$ (mm) are thresholds for $P_i$ at which the value of $f_i$(overcast) changes.


### 2.3. Potential evapotranspiration

Droogers and Allen (2002) compared the data−intensive FAO−56 application of the

Penman−Monteith equation (Allen et al., 1998) for potential evapotranspiration with the less−demanding

equation by Hargreaves (1994):

$$ET_{0,i} = \frac{0.0023 R_{A,i}}{2.501 - 0.001185 \left( T_{\max,i} + T_{\min,i} \right)} \max \left[ \left( \frac{T_{\max,i} + T_{\min,i}}{2} + 17.8 \right), 0 \right] \cdot$$

$$\left( T_{\max,i} - T_{\min,i} \right)^{\frac{1}{2}} \tag{4}$$

where $ET_{0,i}$ (mm) denotes the potential evapotranspiration on day $i$, and $R_{A,i}$ is the extraterrestrial radiation

in MJ m$^{-2}$ d$^{-1}$. The original equation was modified by taking into account Evett's (2000) relationship (his



Eq. 5.45) between the heat of vaporization of water and the temperature. Another modification prevents

the potential evapotranspiration from becoming negative during very cold days. The fact that this is

necessary suggests that the equation should be used with care in cold climates.

Droogers and Allen (2002) also introduced a modified version of the Hargreaves equation that

included a precipitation term. They found good agreement between the Penman−Monteith equation and

the original Hargreaves equation. When inaccuracies were introduced in the meteorological data used in

the calculations, the modified Hargreaves equation outperformed both the Penman−Monteith equation and

the original Hargreaves equation. Here, both versions of the Hargreaves equation are implemented. The

original version only needs the daily solar radiation and the daily minimum and maximum temperature.

The modified version also needs the monthly precipitation. To reduce the occurrence of abrupt changes in

the potential evapotranspiration, the precipitation in the 30−day time window centered on the end of the

day for which the weather is generated is used here instead of its monthly sum, leading to:

$$ET_{0,i} = \frac{0.0013 R_{A,i}}{2.501 - 0.001185\left(T_{\max,i} + T_{\min,i}\right)} \max\left[\left(\frac{T_{\max,i} + T_{\min,i}}{2} + 17.0\right), 0\right] \cdot$$

$$\left\{\max\left[\left(T_{\max,i} - T_{\min,i}\right) - 0.0123 \sum_{j=i-14}^{j=i+15} P_j, 0\right]\right\}^{0.76} \tag{5}$$

Note that the equation was modified from the version of Droogers and Allen (2002) to ensure that

potential evapotranspiration cannot become negative during very cold days or very wet periods. Here too,

the heat of vaporization was made temperature−dependent according to Evett (2000).

The extraterrestrial radiation $R_{a,i}$ depends on the latitude and the time of year. The following

expressions were adopted from Šimůnek et al. (2013, p. 42−43):

$$R_{a,i} = \frac{0.0864 G_{sc}}{\pi}\left\{1 + 0.033 \cos\left[\frac{2\pi(i \bmod 365)}{365}\right]\right\}(\omega_s \sin\zeta \sin\xi + \sin\omega_s \cos\zeta \cos\xi) \tag{6a}$$

with

$$\omega_s = \arccos(-\tan\zeta \tan\xi) \tag{6b}$$

$$\xi = 0.409 \sin\left[\frac{2\pi(i \bmod 365)}{365} - 1.39\right] \tag{6c}$$





Here, $\omega_s$ (rad) is the sunset hour angle, $\xi$ (rad) is the solar declination, and $\zeta$ (rad) is the latitude of the

location of interest. $G_{sc}$ is the solar constant, with values reported as 1353 (Hillel, 1998, p. 595), 1370

(Evett, 2000, p. A−137), and 1360 (Šimůnek et al., 2013, p. 42) Wm$^{-2}$. In this study, the value of 1360

Wm$^{-2}$ is used. A detailed explanation of Eqs. (6a – 6c) is provided by Evett (2000), specifically his Eqs.

5.10, 5.14, 5.17, and 5.19.

It is worth noting that the original Hargreaves equation (Eq. (4)) only responds to a changing

humidity through the effect it might have on the temperature and its daily fluctuations, while the modified

equation (Eq. (5)) has a direct mechanism for humidity to reduce the potential evapotranspiration $ET_0$

through increased rainfall. If the climate becomes warmer and more humid, mean temperature goes up

(increasing $ET_0$) up but daily fluctuations may diminish (decreasing $ET_0$). The combined effect is

illustrated in Table 2. The effect of the daily temperature fluctuation ($T_{max} - T_{min}$) is pronounced in all

cases: a 10°C increase in the range increases the potential evapotranspiration by a factor 1.73 for the

original equation and by a factor 2.30 for the modified equation during a dry period. Sixty mm of rain in a

30−day period increases this factor to 2.50. The response to a 10°C increase in average temperature is

smaller: 1.28 for the original equation as well as the modified equation (irrespective of the amount of

rain).  Using the modified instead of the original Hargreaves equation caused a 16% reduction of $ET_0$ for

a 5°C daily temperature range, an increase of roughly 1% for a 10°C range, and an increase of about 12%

for a 15°C range, with small variations between the different average temperatures. For the modified

equation, 60 mm of rain reduces potential evapotranspiration relative to dry conditions by 4% for a 15°C

daily temperature range and 11% for a 5°C range.


## 3. Generating weather records

        The weather generator first requires the rainfall record. When daily rainfall sums are available for

the full modeling period, the temperature record can be generated. With the daily rainfall sums and the



daily mean, minimum, and maximum temperatures available, the record with daily potential

evapotranspiration can be generated.

If one selects the parameters to generate a weather record with desirable properties that are

defined *a priori*, several attempts will usually be necessary. Generating a rainfall record can be time

consuming, especially for humid climates with a large number of cells. For the temperate climate scenario

described below, generating 100 years of rainfall took 33 minutes on a laptop computer with four Intel

Core i5−3230M 2.60 GHz CPUs. For the arid climate described below, 100 years of rainfall took less than

15 s. The generation of the temperature and evapotranspiration records took a few seconds for a century of

data for either scenario. To ease the testing of different parameter sets, the FORTRAN90 code is therefore

provided as two different programs: one for the rainfall record and the other for the temperature/potential

evapotranspiration record. The rainfall generator's output files Daily_Rain.OUT, Monthly_Rain.OUT, and

Annual_Rain.OUT serve as input for the temperature and evapotranspiration generator. The rainfall

generator code and the temperature/evapotranspiration generating code are provided in the supplement.

When trying out different parameters combinations to generate a rainfall record with desirable

properties, it is recommended for this stage to generate relatively short rainfall records (4 or 8 years for

humid climates, up to 100 years for arid climates) to keep the calculation times short. Once the rainfall

record is judged adequate, a record can be generated for the full period. One can then tune the temperature

parameters by carrying out as many trials as necessary, using the same daily, monthly, and annual rainfall

records on input every time.

The rainfall generator can optionally provide additional information about rain storms and rain

cells, generate a file with the continuous timeline of irregular intervals and their rainfall rates, and

generate a file of hourly rainfall sums. For long records, these options can all be set to 'No' in the input

file RainPar.IN to avoid the generation of very large files. Such records can be generated for a shorter

period if desired. If the seed for the random generator is not changed, the shorter record is identical to the

first years of the long record.



### 3.1. Generating a rainfall record


The time periods for which parameters must be specified can be the months of the year. In areas

with limited data and/or erratic rainfall this will often not be realistic. Therefore, a year can be divided into

an arbitrary number of time periods of arbitrary duration. It is expected that in many regions on the globe,

two periods may often suffice. If a time period extends into the next year it must be split in two: one

ending at December 31$^{st}$, and the other starting at January 1$^{st}$. The last storm generated for a given period

will start in a later period. Its parameters are drawn from distributions valid for the period in which it

occurs.

In order to be able to generate very long records of rainfall, leap years are accounted for. The

model generates rainfall records for a multiple of four years (rounding up when necessary), and the first

leap year occurs in year four. The model determines internally in which period February 28$^{th}$ falls and

assigns February 29$^{th}$ to the same period.

The parameters of Table 1 are required for each of the periods in which a year is divided. The

number and duration of these periods are specified in the input file. A rainfall record is generated by first

determining the starting times and durations of all rain storms for the full duration of the weather record.

Storm start times are generated for a period until the start time falls outside that period. This storm is not

discarded, but its parameters are defined by the period in which it starts and subsequent storm start times

are generated from there.  This implies that storms can overlap, but also that entire periods can be without

rain.

The first rain cell of a storm starts when the storm starts. Subsequent cells in a storm are generated

until the starting time exceeds the end time of the storm. This last cell is discarded. Thus, each storm

consists of at least one cell. For each cell, its duration and rainfall rate are drawn from their respective

distributions. Thus, cells can overlap with the next storm(s), even if storm start and end times are such that

the storms themselves do not. The rainfall rate at any time equals the sum of the rainfall rates of all cells

that are active at that time. From the start and end times of all cells in the rainfall record the complete time

line of rainfall rates and time intervals to which they apply is computed. From this time line, hourly, daily,



monthly, and annual rainfall sums are determined. All records are written to output, as well as the means

and standard deviations of the rainfall for each month and the annual rainfall.

Exponentially distributed random variates with mean $1/\lambda$ are obtained from the standard uniform

variate $U$ through the transformation $-\ln(U) / \lambda$ (Abramowitz and Stegun, 1970, p. 953).

gamma−distributed variates are generated according to the algorithm of Xi et al. (2013).

The following procedure is proposed to determine the model parameters when insufficient data

are available to fit the model:

*1. Define the time periods with different rainfall characteristics into which a year is divided*

*2. For each of these time periods run through the following parameter setting sequence*

a.  Set the mean time interval between storm arrival times. This equals $\lambda^{-1}$.

b.  Set the mean duration of rain cells in a storm. This equals $\nu/\alpha = 1/\bar{\eta}$.

c.  Find a pair $(\alpha, \nu)$ that produces a suitable distribution of $\eta\,(\mathrm{d}^{-1})$. Note that this distribution can be

truncated by setting a minimum value for $\eta$ (see below).

For convective showers, cells will be short−lived so $\eta$ will usually be large. The Gamma

distribution of $\eta$ should have a modest left tail and a heavy right tail. The shape factor $\alpha$ needs to

exceed 1, and probably $\alpha \gg 1$ (suggestion: $> 4$). The rate factor can be $\ll 1$ (suggestion: $< 0.1$).

Consequently, the ratio $\alpha / \nu$, and therefore $\bar{\eta}$, is large.

For frontal rain, the rain cells that together constitute the storm can last long (hours or more). To

avoid excessively long cells one can set $\alpha > 1$, ensuring that the pdf equals zero for $\eta = 0$.

Nevertheless, a minimum value of $\eta$ (denoted $\varepsilon$, see below) may need to be set to some positive

value. The rate factor will have to be high enough to avoid frequent occurrences of large values of

$\eta$ that give short cells. The ratio $\alpha / \nu$, and therefore $\bar{\eta}$, will therefore be small.

The degree of variation in $\eta$ can be modified through $\alpha$ by noting that the coefficient of variation

for the gamma−distribution equals $\alpha^{-1/2}$ (Walpole and Myers, 1978, p. 132). Because $\alpha$ will

generally exceed 1, the coefficient of variation ranges between 0 and 1.



d.  Set the desired mean duration of a storm. This equals $\varphi\alpha / v$, thus defining $\varphi$.

e.  Set a representative time interval between the starting times of the rain cells within an individual storm. This interval equals $1/\kappa\bar{\eta}$, and therefore defines $\kappa$. The larger $\kappa$ is, the more likely it is that cells overlap. This can be used to generate short periods of high rainfall rates within a storm. This should be taken into account when setting the parameters that govern the rainfall rate of rain cells (see below).

    The number $1 + \kappa /\varphi$ is roughly representative of the mean number of cells in a storm. For many cases, a target ratio of $\kappa /\varphi$ between 3 and 6 seems to work well. Large ratios lead to large numbers of cells and can cause excessive calculation times.

f.  Set the mean rainfall rate of a rain cell. This equals $p /\delta$ or $m^{-1}$. For large values of $\kappa$, excessive tailing (with high rainfall rates) is undesirable and the rate factor $\delta$ should be high, unless short−lived very high rainfall rates caused by multiple overlapping cells are desired. Here too, the degree of variation can be tuned by selecting $p$ such that the coefficient of variation ($p^{-1/2}$) has the desired value.

g.  The truncation parameter $\varepsilon$ is the minimum permissible value for $\eta$. It can be determined by setting a critical duration of a rain cell $t_{crit}$ (T) and a permissible probability $f_{crit}$ that the duration of a cell exceeds $t_{crit}$. From the cell duration's cumulative pdf with parameter $\eta$ follows:

$$1 - (1 - e^{-\eta t_{crit}}) = f_{crit} \tag{7}$$

Solving for $\eta$ gives its critical value. This is the desired value of $\varepsilon$.

$$\varepsilon = -\frac{\ln(f_{crit})}{t_{crit}} \tag{8}$$

If the following condition is met truncation is not required, and $\varepsilon$ can be set to zero.

$$\left(\frac{v}{v+t_{crit}}\right)^{\alpha} < f_{crit} \tag{9}$$

The appendix shows the derivation of Eq. (9).





h.  When the truncation of $\eta$ is ignored, the average total rainfall during the period can be estimated

using the averages of the number of storms, storm duration, number of cells per storm, cell

duration, and cell rainfall rate. The average number of storms in an $n$–day period equals $n / \lambda$.

The estimated average total rainfall $P_{tot}$ (mm) of the period is given as

$$P_{tot} = \frac{nvp}{\lambda\alpha\delta}\left(\frac{\varphi}{\kappa} + 1\right)$$

With this equation, available data for average monthly rainfall can thus be used to further

constrain the model parameters early in the iterative process.

The input requirements are detailed in the User Manual in the supplement, as well as in the heading of the

rainfall generating code (WhollStopTheRain3.F90) in the supplement.

### 3.2. Generating a temperature and evapotranspiration record

Setting the time to zero at 0:00 hrs. on January first of the first year after a leap year allows day

1155 (February 29[th]) of each subsequent four–year period of 1461 days to be given the same value for the

argument of the sine function in Eq. (1a) as February 28[th], and the remaining days of that leap year will be

treated as if they are in a non–leap year. This ensures that the time axis of the temperature record is

synchronized with that of the rainfall record. The code implements this by using the output files with the

daily, monthly, and annual rainfall sums generated by the rainfall as input.

To determine whether day $i$ is cloudy, $P_i$ is taken from the rainfall record, $f_i$(overcast) determined

accordingly from Eq. (3), and a random value $u$ of a standard uniform variate $U$ selected. If $u <$

$f_i$(overcast), the day is overcast. In that case, $A_o$ and $\sigma_{f,o}$ are used in Eqs. (1a) and (2) to generate the

various temperatures for day $i$. Otherwise, $A_c$ and $\sigma_{f,c}$ are used. The standard normally distributed variates

necessary for fluctuations of the mean temperature and the daily range are generated from two standard

uniformly distributed variates $U_1$ and $U_2$ as $\cos(2\pi U_1)\sqrt{-2\ln U_2}$ (Press et al., 1992, p. 279). If the



dependence between daily temperatures and daily rainfall is undesired, in the input file, $A_o$ can be set

equal to $A_c$, and $\sigma_{f,0}$ equal to $\sigma_{f,c}$. In that case the parameters in Eq. (3) can be given arbitrary values.

If local meteorological data are not available, the shift and average amplitude of the annual

temperature signal can be estimated from data that several websites for weather information provide for

locations worldwide. These websites often also provide monthly averages of minimum and maximum

temperatures. These data can serve as a starting point for estimating the parameters that govern the daily

variations of the mean and extreme temperatures. The estimation of the parameters of the lognormal

distribution governing the daily temperature extremes is facilitated by a worksheet that either allows the

user to calculate directly exceedance probabilities for selected values of the difference between the daily

maximum and mean temperature for a proposed set of input parameters $\mu_f$ and $\sigma_{f,n}$ in Eq. (2). Alternatively

this is done for user−supplied values of the mean and standard deviation of the lognormal distribution in

Eq. (2). The corresponding input parameters ($\mu_f$ and $\sigma_{f,n}$) for the temperature generator are then computed

by the worksheet.

Information about the autoregressive nature of daily temperature fluctuations is limited (examples

are the studies by Kalvová and Nemešová, 1998, and Breinl et al., 2015). Furthermore, if values of the

autoregressive coefficient $\phi$ reported in the literature are based on the temperature record in isolation (as

in the case of Kalvová and Nemešová, 1998) they may be in the high end of the range because the weather

generator described here introduces an autoregressive element through the dependence of the temperature

on cloud cover, which in turn is affected by the daily rainfall. If rain occurs in multiday clusters, the

temperature record can be expected to reflect this, even if $\phi$ is set to zero.

The parameters that govern the interannual variation ($\sigma_a$, $\sigma_T$) can only reliably be determined if

long−term records exist. If desired, they can be set to zero to keep the long−term mean temperature and

the amplitude of the sinusoidal annual trend constant.

The calculation of the potential evapotranspiration can proceed after the rainfall record and the

temperature record have been generated. If the modified Hargreaves equation is used, the rainfall over the



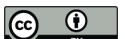

30−day period during the first 14 days is completed by taking the final 14 days (for day 1) to the final day (for day 14) of the rainfall record to replace the missing leading days in the rainfall record. For the final 15 days, the missing trailing days are similarly replaced by the first days of the rainfall record.

The evapotranspiration model does not have any fitting parameters. The only input parameter required in addition to the rainfall and temperature records is the geographic latitude of the location of interest. For this reason, the temperature and evapotranspiration records are generated by a single code (TheHeatIsOn2.F90). The input requirements are detailed in the heading of this code and in the User Manual in the supplement.


## 4. Example calculations

In order to generate a weather record, input files need to be provided for the rainfall generator (RainPar.IN), and the temperature/evapotranspiration generator (TempEpot.IN). Test cases for a temperate and an arid climate were developed. Their input files are given in the supplement. The parameter values

are given in Tables 3−5. One hundred years of weather data were generated.

The rainfall record of the temperate climate was based on the monthly parameter values fitted by Pham et al. (2013) to rainfall data of Uccle (Belgium) that exhibited moderate seasonality. The latitude (0.8866 rad, 50.8°N) also was that of Uccle. The values of $\overline{T}_a$, $A_c$, and $\psi$ were based on the record of average monthly temperatures in Uccle (Meteoblue.com, 2018). The rainfall record of the arid climate was

not based on an existing climate but designed to have two contrasting seasons. The latitude and temperature parameter values are representative of conditions in the Sahara.

### 4.1. Temperate climate

The average annual and monthly rainfall is given in Table 6. The coefficient of variation of the

annual rainfall is 14%. Rainfall in winter is more plentiful and more variable than in summer. The average monthly values range between 21 and 65 mm. Figure 2 shows the daily rainfall of the first four years of



the temperate climate. Prolonged dry periods are absent, and 4 days have 15 mm of rainfall or more (3.1% of the average annual rainfall).

The temperature signal of the first four years (Fig. 3) shows a clear sinusoidal trend. A few days with average below 0°C are common in winter. Rarely does the mean daily temperature rise above 25°C. The daily extremes keep most summer nights below 20°C and give frequent occurrence of frost in winter nights. For most winter days, temperatures rise above zero for some time during the day. Summer daily maximum temperatures rise above 25°C for a few days every year except year 3, but rarely above 30°C.

The difference between the number of daylight hours in summer and winter at the latitude of Uccle causes a large spread in the daily values of the extraterrestrial radiation (Fig. 4). The relatively cool climate gives much lower values of the potential evapotranspiration. Figure 4 shows that the potential evapotranspiration according to Hargreaves (1994; Eq. (4)) tends to be higher than that of Droogers and Allen (2002; Eq. (5)).

The monthly rainfall for the full 100−year record (Fig. 5) confirms the absence of dry periods, and exhibits relatively few extreme outliers: only 2% of the months have more than 100 mm of rainfall. The annual rainfall in Fig. 6 is even more evenly distributed. Potential evapotranspiration rates are consistently higher than annual rainfall. The difference between the values according to Hargreaves (1994) and Droogers and Allen (2002) confirms the differences between their daily values in Fig. 4. Not surprisingly, most years have an evapotranspiration deficit, calculated as the annual sum of the daily difference between rainfall and potential evapotranspiration. This deficit is always larger for Hargreaves (1994) than for Droogers and Allen (2002).

Figure 7 shows the average trends over the year of the temperature and its daily extremes, the extraterrestrial radiation, and the potential evapotranspiration. The lag between the radiation and the temperature is clear. The potential evapotranspiration is influenced by both, and hence its peak lies between that of the radiation and the temperature.





The correlation between average monthly rainfall, cloudiness, and the evapotranspiration surplus/deficit is demonstrated in Fig. 8. The evapotranspiration deficit occurs in the period May − September, covering most of the growing season in temperate climate zones. In this period, the difference between the deficits calculated according to Hargreaves (1994) and Allen and Droogers (2002) is the

largest.

### 4.2. Arid climate

Table 6 shows that the rainfall is almost entirely concentrated in the period November − January. The variability is very high, with coefficients in variation larger than 100% for all months except January.

As a result, the coefficient of variation of the annual rainfall is much larger than that of the moderate climate: 71%. The average monthly values for the dry season range between 0.5 and 6.5 mm, for the wet season between 11 and 27 mm. The daily rainfall of the first four years (Fig. 9) illustrates the massive variation between years. The wettest day of the period delivers about 5% of the average annual rainfall. The aridity of the dry season results in the regular occurrence of dry periods of several months.

Average temperatures range from about 5°C to the nearly 40°C (Fig. 10). The daily range is large: winter maxima frequently are above 15°C, and occasionally above 20°C. Winter minima can be above 10°C but also close to 0°C. Summer nights are above 25°C most of the time. Summer maxima are rarely below 35°C, and can be well over 40°C. The large annual fluctuation gives rise to rapid temperature rises in spring and fast drops in autumn.

The location is relatively close to the equator (latitude 0.3977 rad, corresponding to 22.8°N). The annual fluctuation in the extraterrestrial radiation (Fig. 11) is therefore much smaller than in Uccle (Fig. 4). The warm weather and large daily temperature fluctuations give rise to high potential evapotranspiration rates in summer. The day−to−day variation of the potential evapotranspiration is considerable, but the systematic difference between the values according to Eqs. (4) and (5) appears to be

much smaller than that for Uccle.



Figure 12 shows the monthly rainfall for the arid climate. The arid periods separating the wet

seasons are clearly apparent, including 21 prolonged droughts with at least 5 consecutive months that have

less than $5 \cdot 10^{-5}$ mm of rainfall. One of those droughts spans the entire nine−month dry season (months

899−907). The wet months have widely varying rainfall amounts. Eighteen months have more rainfall

than the annual average. The wettest of them (month 839) has 146 mm, more than twice the average

annual rainfall.

The annual rainfall (Fig. 13) is also highly variable. The wettest year (82) had 95 times as much

rainfall as the driest (year 93). The potential evapotranspiration is nearly 20 times larger than the average

annual rainfall, leading to evapotranspiration deficits of well over 1 m. As is the case for the temperate

climate, Hargreaves (1994) gives larger evapotranspiration deficits than Droogers and Allen (2002), but

the difference is roughly half as large.

Figure 14 shows a relatively flat annual trend of the extraterrestrial radiation, in line with the

limited seasonality near the equator. In combination with the pronounced temperature trend this leads to a

trend in the potential evapotranspiration that is nearly synchronized with the temperature. Droogers and

Allen (2002) gives only slightly lower values of the potential evapotranspiration than Hargreaves (1994),

but does so consistently through the year.

Figure 15 clearly show the seasonality in the rainfall and the cloudiness that the arid climate was

designed to demonstrate. There is a permanent and large evapotranspiration deficit. The difference

between the deficits according to Hargreaves (1994) and Allen and Droogers (2002) is insignificant in

comparison.

Table 7 allows a comparison between weather characteristics of both climates. The increase of

40% in potential evapotranspiration from the temperate to the arid climate leads to an increase in the

potential evapotranspiration of 117% (Eq. (4) − Hargreaves, 1994) or 145% (Eq. (5) − Droogers and

Allen, 2002). The evapotranspiration deficit increases by a factor 9 (Eq. (4)) or a factor 20 (Eq. (5)).

The potential evapotranspiration and its deficit according to Eqs. (4) and (5) differ strongest in

both absolute and relative terms for the temperate climate. For the arid climate, the difference is 30 mm



for both, a difference of less than 2.4%. For the temperate climate, the difference is 85 mm for both, which

constitutes a 16% difference for the potential evapotranspiration and a 132% difference for the

evapotranspiration deficit, when all relative differences are calculated with reference to the values based

on Droogers and Allen (2002) − Eq. (5). For both climates, Droogers and Allen (2002) give the smaller

estimate of the potential evapotranspiration, and consequently also the smallest evapotranspiration deficit.

For temperate climates, the differences are large enough to merit careful consideration. Table 7

demonstrates the enormous effect the choice of the evapotranspiration equation can have. Table 2 shows

that not only the average daily temperature but also the difference between the daily temperature extremes

has a large effect on the potential evapotranspiration of that day. This finding draws attention to the

importance of the temperature model, including the model for the daily temperature range.

### 5. Concluding remarks

A relatively simple weather generator was presented that can produce very long records of hourly

and daily rainfall (and aggregated data), daily values of the average, minimum, and maximum

temperatures, daily values of the potential evapotranspiration according to two models, and daily binary

cloud−cover (complete cloud cover or clear skies). The weather generator's simplicity facilitates the

'construction' of desired climates by choosing the appropriate parameter values, for which a procedure

was proposed.

The weather generator was used to generate 100 years of data for two very contrasting climates. In

doing so, the two evapotranspiration models were shown to give comparable evapotranspiration rates and

deficits for an arid climate, but very different values for a temperate climate. Further analysis showed that

not only the daily mean temperature but also the difference between the daily extremes strongly affects the

daily evapotranspiration rate within each evapotranspiration model.

The rainfall generator is independent of the other modules of the weather generator. To facilitate

trial−and−error parameter estimation and more formal calibration procedures, the rainfall generator is



presented as a separate code. Three of its output files are used on input by the code that generates the

remaining weather variables.

Both codes are written in Fortran−90 and were compiled with the GNU Fortran compiler

(gcc.gnu.org) operated through the CygWin interface (cygwin.com) to allow it to run under the Windows

operating system. The executables run on standard laptop and personal computers. The stand−alone nature

of the codes (without the need for commercial software packages or heavy computational capacity) and

their compatibility with a public domain compiler ensure that this weather generator can be deployed in

areas with limited resources. To further enhance its usefulness, the weather generator produces data that

are required on input by another public domain code, the soil−plant model Hydrus−1D (Šimůnek et al.,

2013, 2016, pc−progress.com), which can use the rainfall and temperature records to calculate crop

growth, and provide information that can be used to schedule irrigations. Hydrus−1D internally uses a

simpler version of Eq. (4), without the temperature−dependence of the heat of vaporization, to calculate

potential evapotranspiration. The weather generator can be used to determine if Eq. (5) would give

significantly different results. If desired, the evapotranspiration record generated by either Eq. (4) or Eq.

(5) can be supplied to Hydrus−1D on input.

**6. Code availability, hardware requirements, and license information**

The source code, executables, example input files, and the spreadsheet file to aid in the selection

of the parameters of Eq. (2) can be downloaded free of cost from

https://www.ufz.de/index.php?en=44055. The codes do not require any special hardware beyond a

standard laptop or personal computer, but 64−bit processors are recommended. The spreadsheet that helps

determine the values of the parameters of Eq. (2) is in Excel.

Users can use and modify the codes as they wish, provided they give prominent notice that they

used these codes and include a proper reference to this paper. The source code, executables, and all other

files come without any warranty, including any implied warranty of merchantability or fitness for a

particular purpose. Neither the author nor the Helmholtz Zentrum für Umweltforschung – UFZ, GmbH





can be held liable for any consequence from the use of the codes and/or files presented here or any

modifications thereof.




**Appendix: Derivation of a criterion that can be used to assess the need to truncate the distribution of $\eta$**

The probability density function (pdf) of the cell duration $y$ (T) depends on $\eta$:

$$f_y(y)\big|_\eta = \eta e^{-\eta y} \tag{A1}$$

The parameter $\eta$ is gamma−distributed with shape parameter $\alpha$ and rate parameter $v$:

$$f_\eta(\eta)\big|_{\alpha,v} = \frac{v^\alpha}{\Gamma(\alpha)}\eta^{\alpha-1}e^{-v\eta} \tag{A2}$$

To find the pdf of $y$ unconditional on $\eta$, we need to integrate over all possible values of $\eta$ while accounting for the pdf of $\eta$:

$$f_y(y)\big|_{\alpha,v} = \int_0^\infty f_y(y)\big|_\eta f_\eta(\eta)\big|_{\alpha,v}\,\mathrm{d}\eta = \frac{v^\alpha}{\Gamma(\alpha)}\int_0^\infty \eta^\alpha e^{-(y+v)\eta}\,\mathrm{d}\eta = \frac{\alpha v^\alpha}{(y+v)^{\alpha+1}} \tag{A3}$$

The integral was evaluated according to Harris and Stocker (1998, p. 992) and the relationship $\Gamma(\alpha+1)/\Gamma(\alpha) = \alpha$ was invoked (Abramowitz and Stegun, 1970, p. 256, Eq. (6.1.15)).

If the range of $\eta$ is truncated by imposing a minimum value $\varepsilon$, the lower bound of the integral changes and the result must be scaled to ensure the resulting pdf integrates to 1. We first define the

truncated gamma distribution:

$$f_\eta(\eta)\big|_{\alpha,v,\varepsilon} = \frac{\frac{v^\alpha}{\Gamma(\alpha)}\eta^{\alpha-1}e^{-v\eta}}{\int_\varepsilon^\infty \frac{v^\alpha}{\Gamma(\alpha)}\eta^{\alpha-1}e^{-v\eta}\,\mathrm{d}\eta} = \frac{\eta^{\alpha-1}e^{-v\eta}}{\int_\varepsilon^\infty \eta^{\alpha-1}e^{-v\eta}\,\mathrm{d}\eta} \tag{A4}$$

The integral can be evaluated if we replace the exponential function by a rapidly converging power series (Harris and Stocker, 1998, p. 234).

$$f_\eta(\eta)\big|_{\alpha,v,\varepsilon} = \frac{\eta^{\alpha-1}e^{-v\eta}}{\int_\varepsilon^\infty \eta^{\alpha-1}e^{-v\eta}\,\mathrm{d}\eta} = \frac{\eta^{\alpha-1}e^{-v\eta}}{\sum_{j=0}^\infty \int_\varepsilon^\infty \eta^{\alpha-1}\frac{(-v\eta)^j}{j!}\,\mathrm{d}\eta} = \frac{\eta^{\alpha-1}e^{-v\eta}}{\sum_{j=0}^\infty (-1)^j \left[\frac{v^j\eta^{\alpha+j}}{(\alpha+j)j!}\right]_\varepsilon^\infty} \tag{A5}$$

Note that the denominator is a constant. The pdf of $y$ conditional on $\alpha$, $v$, and $\varepsilon$ is:

$$f_y(y)\big|_{\alpha,v,\varepsilon} = \int_\varepsilon^\infty f_y(y)\big|_\eta f_\eta(\eta)\big|_{\alpha,v,\varepsilon}\,\mathrm{d}\eta = \frac{\int_\varepsilon^\infty \eta^\alpha e^{-(y+v)\eta}\,\mathrm{d}\eta}{\sum_{j=0}^\infty (-1)^j \left[\frac{v^j\eta^{\alpha+j}}{(\alpha+j)j!}\right]_\varepsilon^\infty} = \frac{\sum_{j=0}^\infty (-1)^j \left[\frac{(y+v)^j\eta^{\alpha+j+1}}{(\alpha+j+1)j!}\right]_\varepsilon^\infty}{\sum_{j=0}^\infty (-1)^j \left[\frac{v^j\eta^{\alpha+j}}{(\alpha+j)j!}\right]_\varepsilon^\infty} \tag{A6}$$



Breaking off the infinite series after the first term leads to a mathematically unacceptable result. We
therefore have to use the untruncated version (Eq. (A3)), even though this means it is not possible to
define a desirable value for $\varepsilon$ that takes into account the pdf of $\eta$.

We want to avoid that the probability of $y$ exceeding a critical cell duration $t_{crit}$ becomes larger
than $f_{crit}$. We therefore need to find the cumulative distribution function of $y$, denoted $F_y(y)$. From Eq. (A3)
we find:

$$F_y(y)\big|_{\alpha,\nu} = \alpha\nu^\alpha \int_0^y (\xi + \nu)^{-\alpha-1}\mathrm{d}\xi = \left[-\left(\frac{\nu}{\nu+\xi}\right)^\alpha\right]_0^y = 1 - \left(\frac{\nu}{\nu+y}\right)^\alpha \qquad (A7)$$

where $\xi$ is an integration variable. With Eq. (A7) we can formalize the limit on the exceedance probability

as:

$$\left(\frac{\nu}{\nu+t_{crit}}\right)^\alpha < f_{crit} \qquad (A8)$$

If the inequality holds for given values of $\alpha$, $\nu$, $t_{crit}$, and $f_{crit}$ , the gamma distribution does not need to be
truncated, and $\varepsilon$ can be set to zero.

The left hand side of the inequality (A8) is the probability that $t_{crit}$ is exceeded by a rain cell when

the gamma distribution of $\eta$ is not truncated. The probability of large cell durations is small if $t_{crit} \gg \nu$.
Large values of $\alpha$ further decrease the occurrence of long cells. If, on the other hand, $t_{crit} \ll \nu$, the
probability that $t_{crit}$ is exceeded approximates 1, implying that only values of $\eta$ from the far end of the
right tail of the distribution are permitted. In this case, either the gamma distribution of $\eta$ and/or its
parameter values are poorly suited for the climate of interest, or the selected value of $t_{crit}$ is inadequate.

Finally, by setting $t_{crit}$ and $f_{crit}$ to predefined values and adding one additional constraint on $\alpha$ and
$\nu$, one can select their values to ensure that truncation is not necessary. If we prescribe a desired mean
value for $\eta$ we have

$$\bar{\eta} = \frac{\alpha}{\nu} \qquad (A9)$$

Equations (A8) and (A9) lead to an inequality in which $\nu$ is the only unknown:




$$\left(\frac{v}{v+t_{crit}}\right)^{\overline{\eta}v} < f_{crit} \qquad (A10)$$

This inequality can be solved by iteration, providing a range of permissible values for $v$. Equation (A9)

can then be used to find corresponding values of $\alpha$. If instead of the mean, the coefficient of variation

(CV) (standard deviation divided by the mean) is prescribed, we have

$$CV = \alpha^{-\frac{1}{2}} \qquad (A11)$$

This sets the value of $\alpha$. Combining Eq. (A11) with Eq. (A8) leads to the permissible range of $v$:

$$0 < v < \frac{(f_{crit})^{CV^2} t_{crit}}{1-(f_{crit})^{CV^2}} \qquad (A12)$$





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





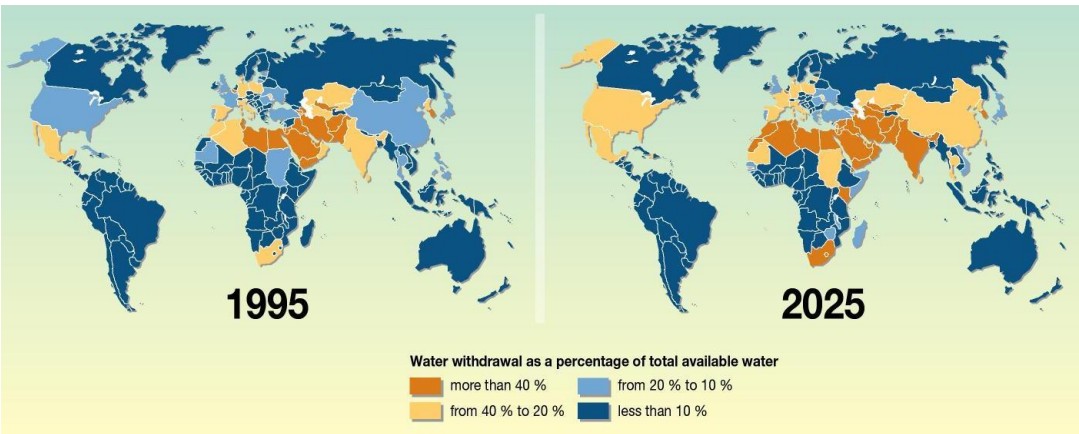

Figure 1. Freshwater use relative to availability per country. Source: P. Rekacewicz, 2006,

http://www.grida.no/resources/5625 (accessed January 11[th], 2018), based on data from Aquastat (FAO,

2016).





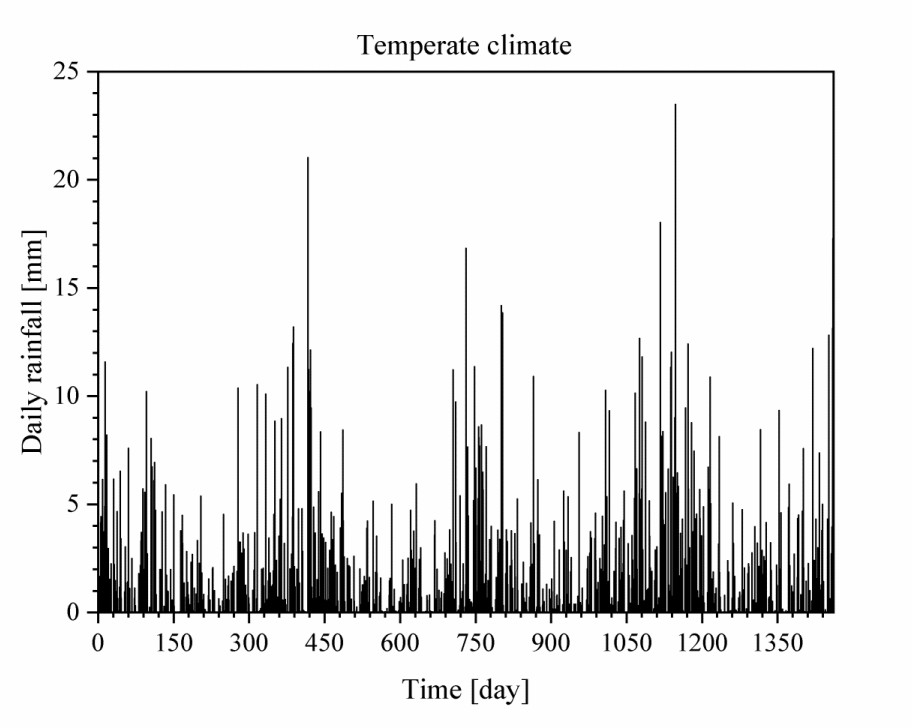

Figure 2. Daily rainfall for a four−year period in a temperate climate. The parameters are given in Table 3.





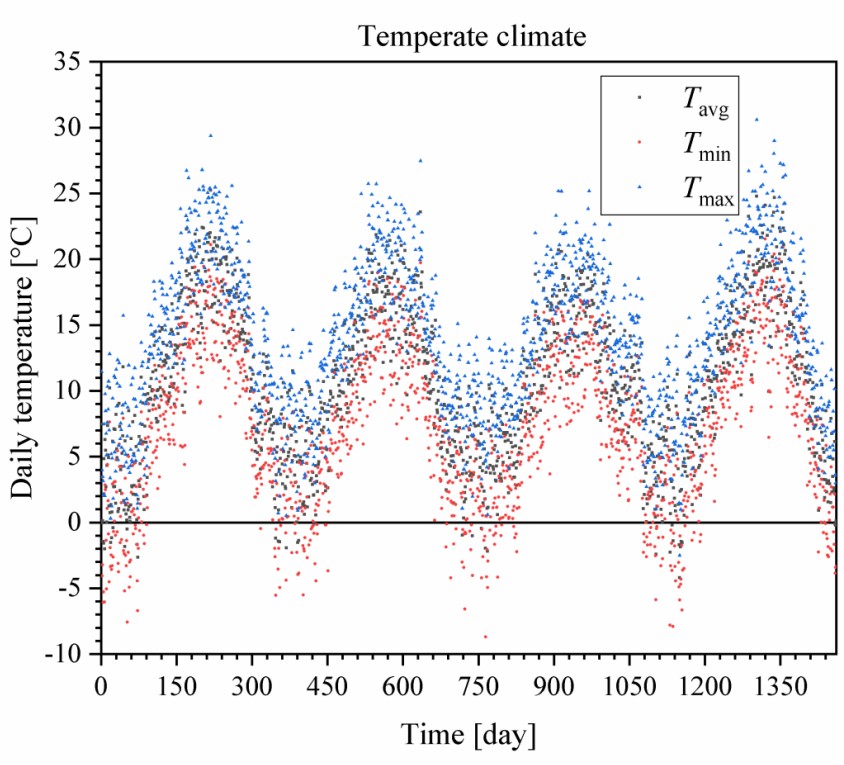

Figure 3. Daily mean, minimum, and maximum temperature for a four−year period in a temperate climate.

The parameters are given in Table 5.






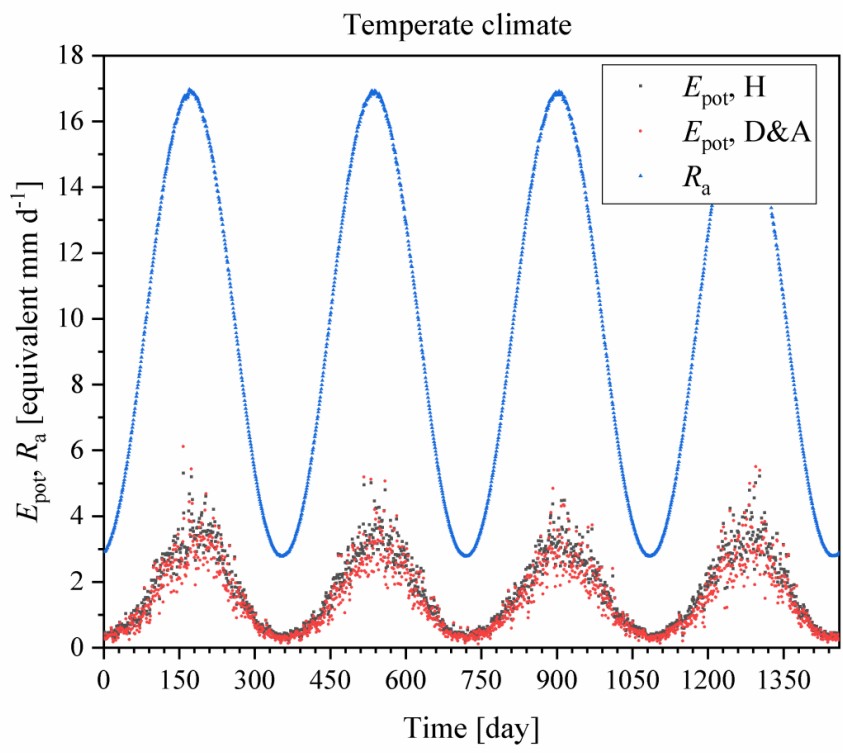

Figure 4. Daily extraterrestrial radiation ($R_a$) and potential evapotranspiration according to Eq. (4) ($E_{pot}$, H) and according to Eq. (5) ($E_{pot}$, D&A) for a four−year period in a temperate climate. The parameters are given in Table 5.






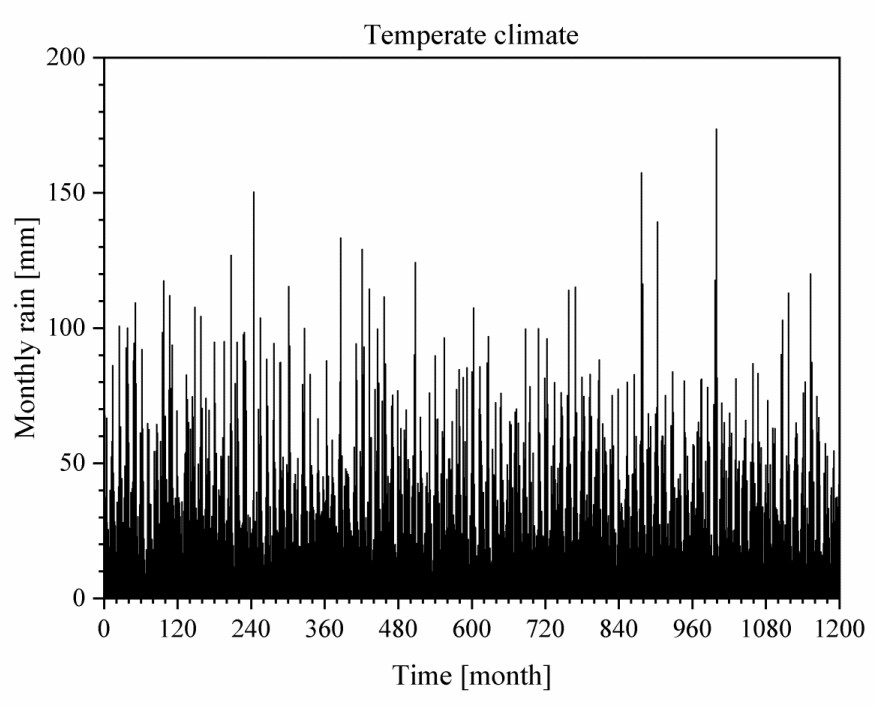

Figure 5. Monthly rainfall for a 100−year period in a temperate climate. The parameters are given in Table 3.



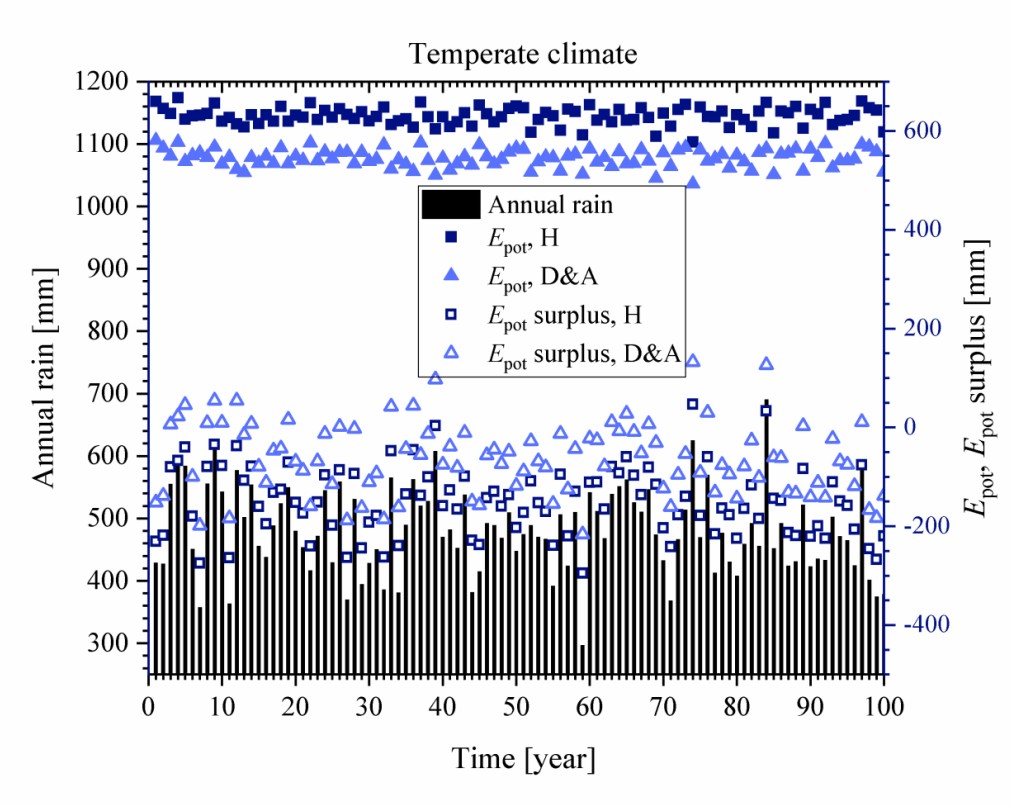


Figure 6. Annual rainfall, potential evapotranspiration according to Eq. (4) ($E_{pot}$, H) and Eq. (5) ($E_{pot}$, D&A), and the corresponding surplus (+) or deficit (−) of the potential evapotranspiration for a 100−year period in a temperate climate.



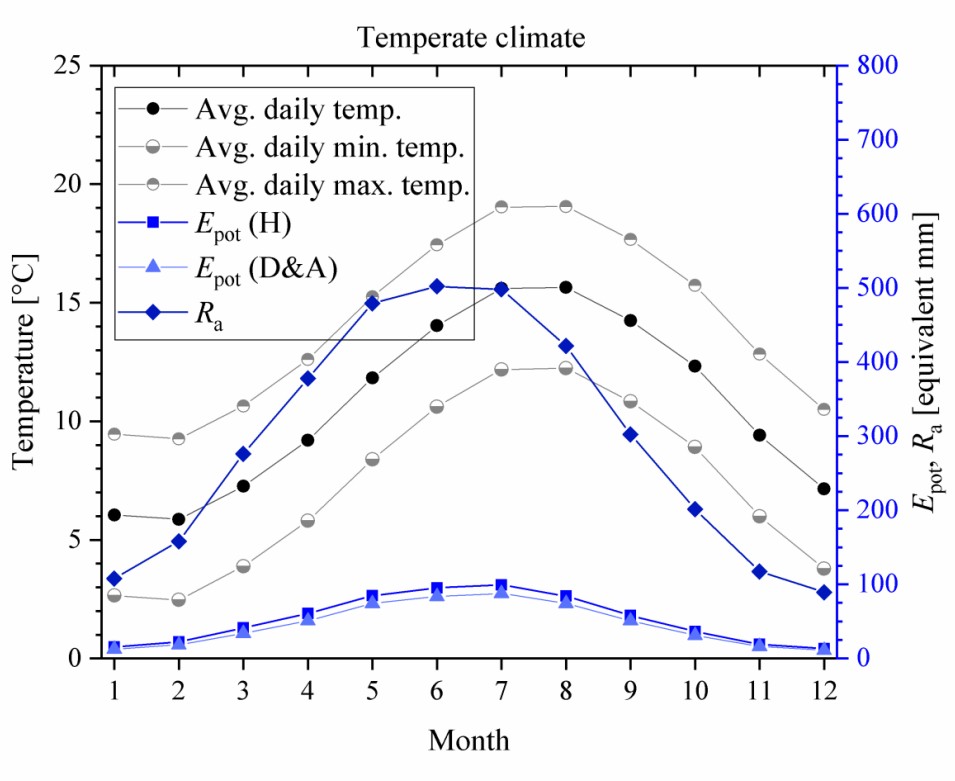


Figure 7. Average monthly values of the daily average, minimum, and maximum temperature, of the extraterrestrial radiation ($R_a$), and of the potential evapotranspiration according to Eq. (4) ($E_{pot}$, H) and Eq. (5) ($E_{pot}$, D&A). The averages are for a 100−year period in a temperate climate.



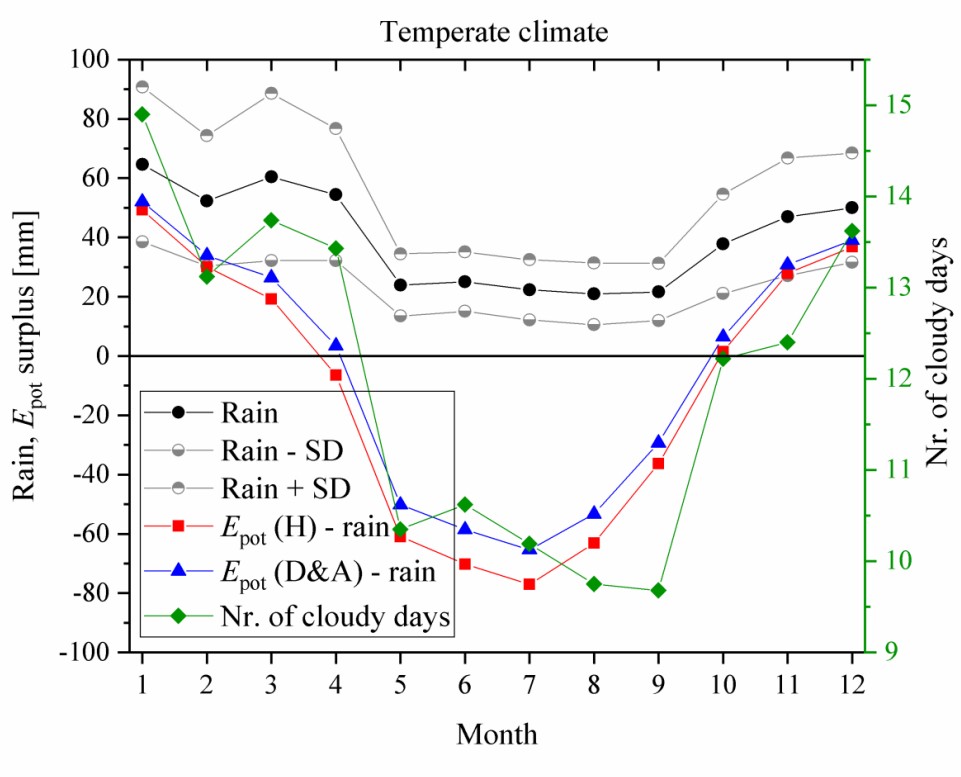


Figure 8. Average monthly values of the monthly rainfall, the number of cloudy days per month, and the monthly evapotranspiration surplus (+) or deficit (−) according to Eq. (4) ($E_{pot}$, H − rain) and Eq. (5) ($E_{pot}$, D&A − rain). The averages are for a 100−year period in a temperate climate.





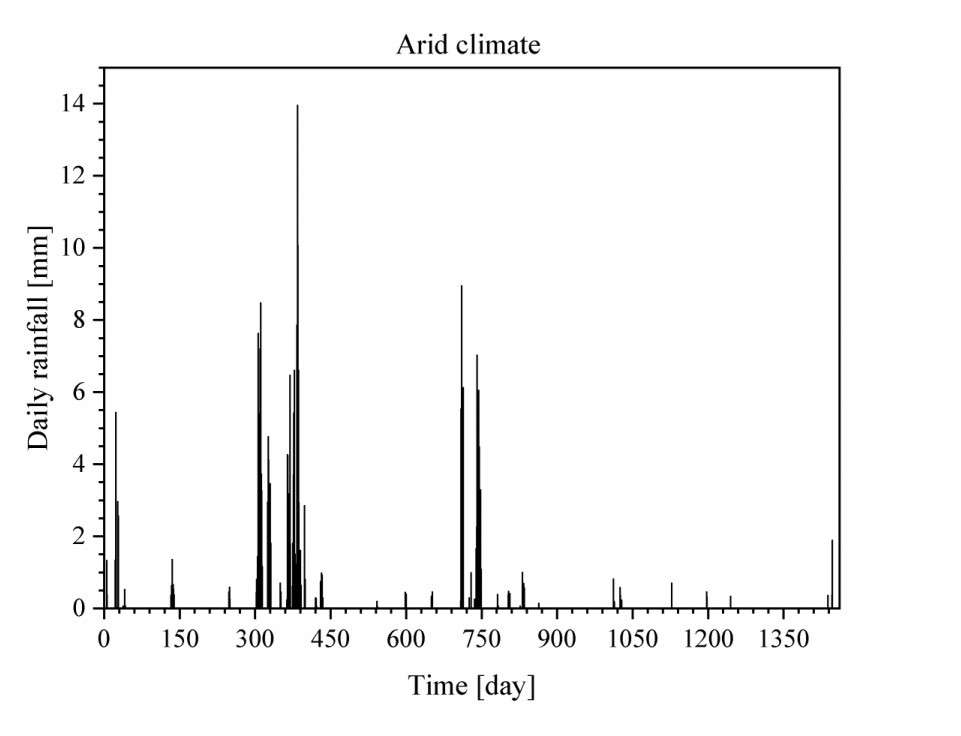


Figure 9. Daily rainfall for a four−year period in an arid climate. The parameters are given in Table 4.





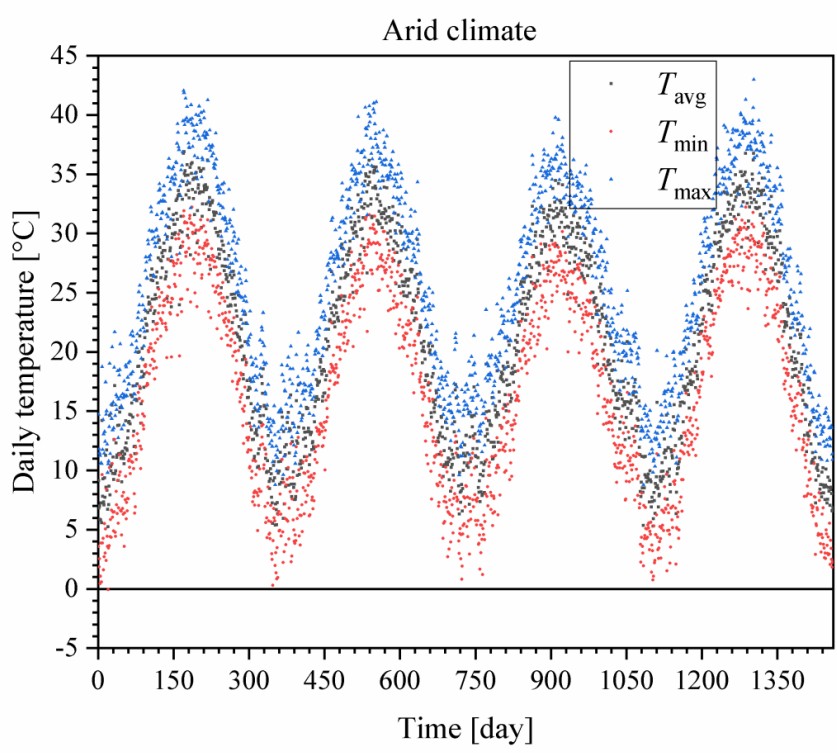

Figure 10. Daily mean, minimum, and maximum temperature for a four−year period in an arid climate.

The parameters are given in Table 5.



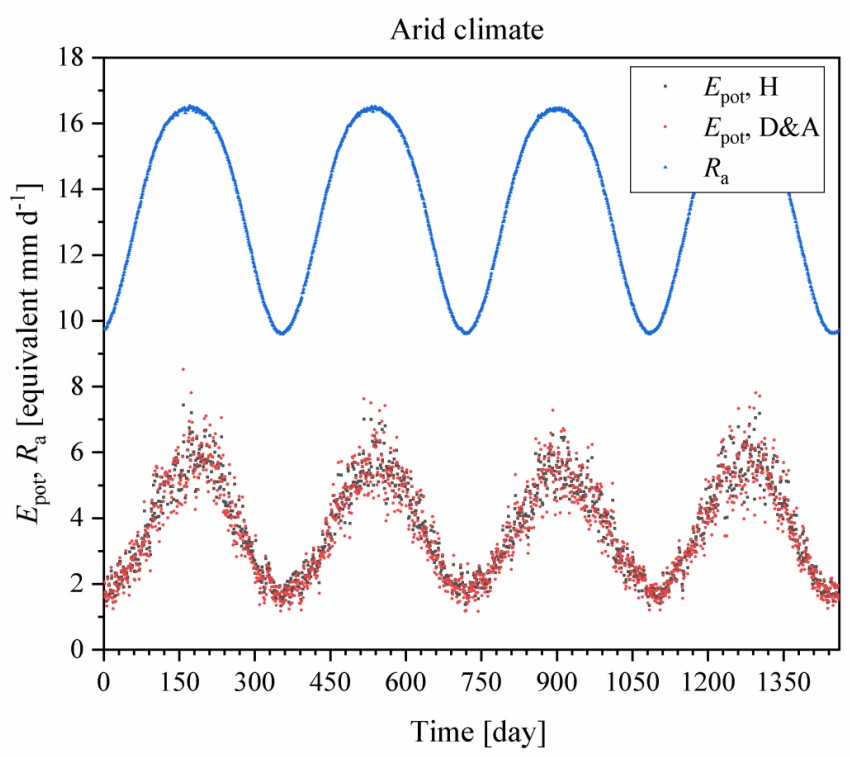

Figure 11. Daily extraterrestrial radiation ($R_a$) and potential evapotranspiration according to Eq. (4) ($E_{pot}$, H) and according to Eq. (5) ($E_{pot}$, D&A) for a four−year period in an arid climate. The parameters are

given in Table 5.





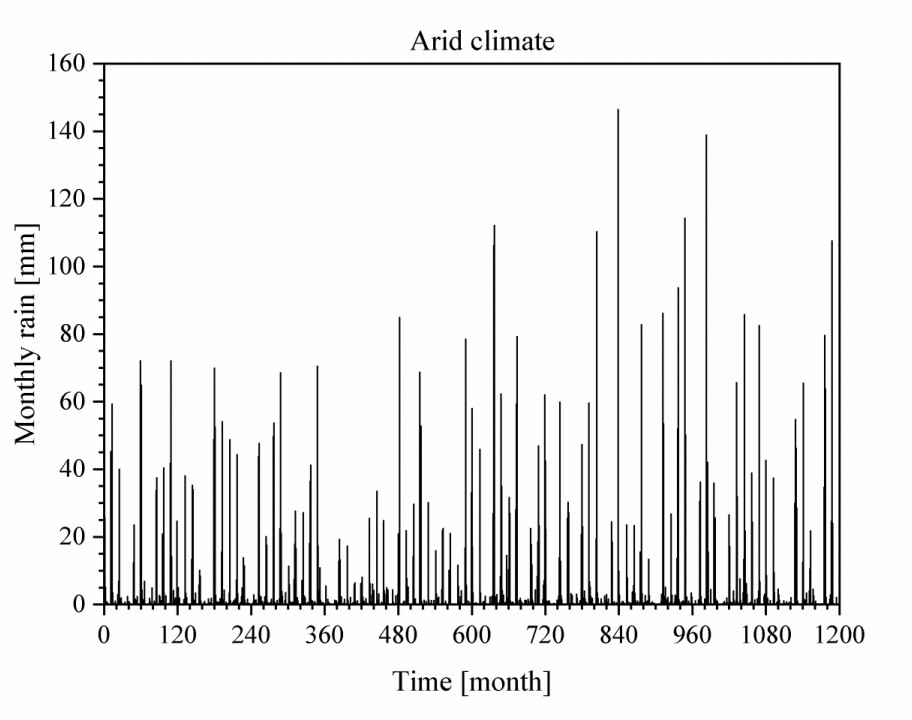

Figure 12. Monthly rainfall for a 100−year period in an arid climate. The parameters are given in Table 4.






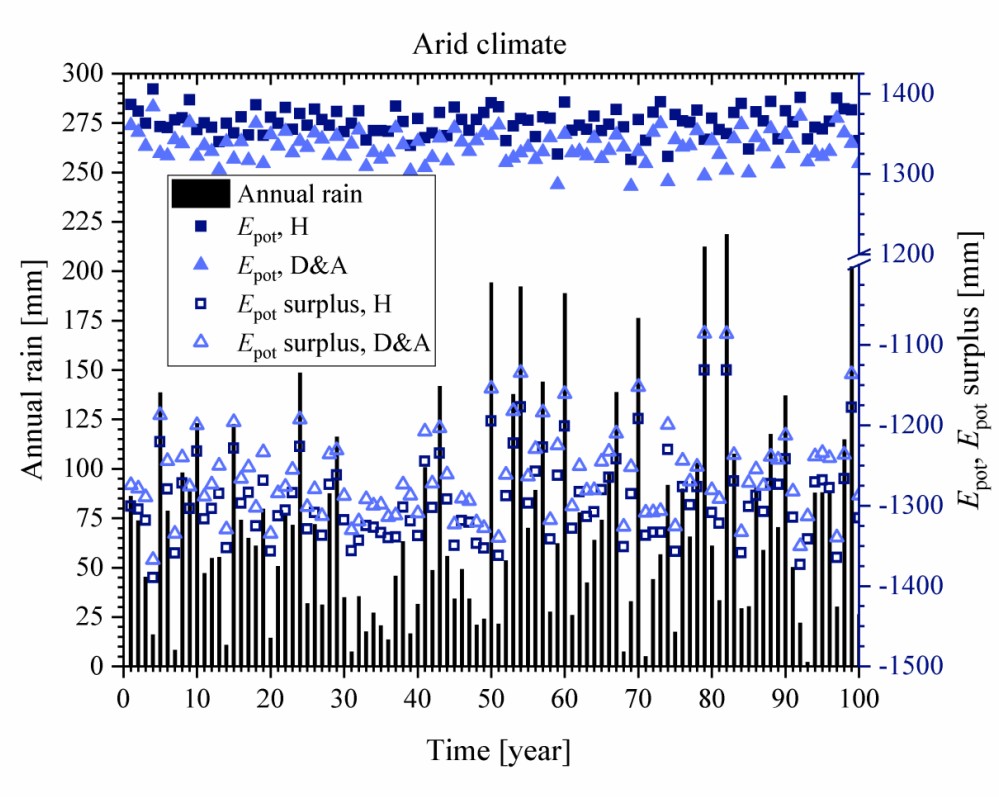

Figure 13. Annual rainfall, potential evapotranspiration according to Eq. (4) ($E_{pot}$, H) and Eq. (5) ($E_{pot}$, D&A), and the corresponding surplus (+) or deficit (−) of the potential evapotranspiration for a 100−year period in an arid climate.





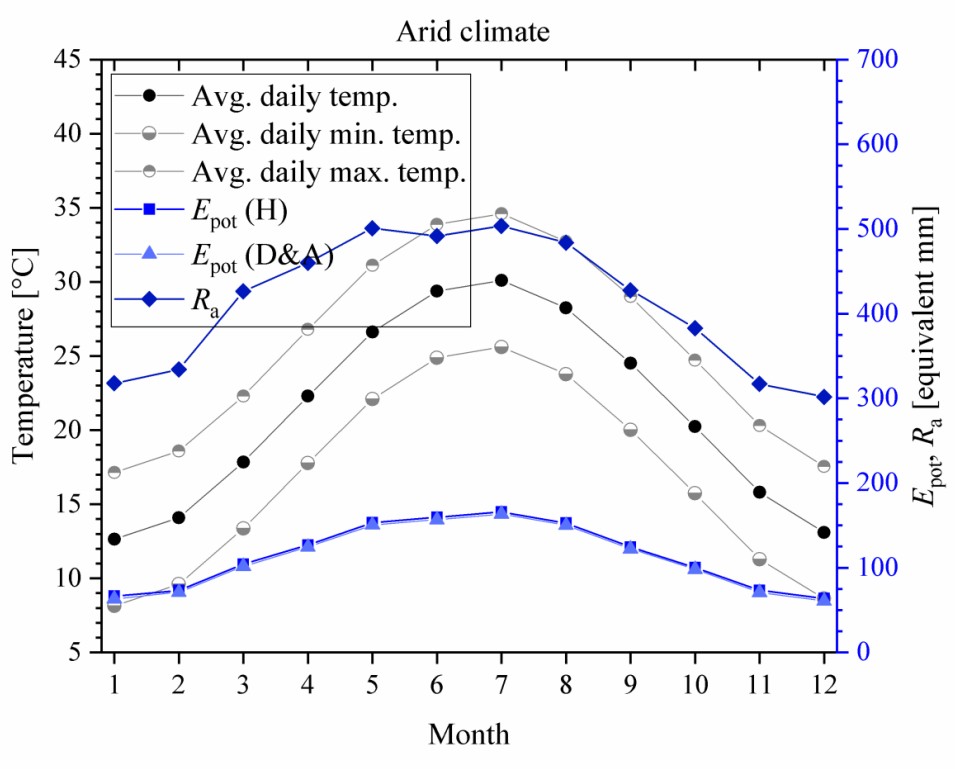


Figure 14. Average monthly values of the daily average, minimum, and maximum temperature, of the extraterrestrial radiation ($R_a$), and of the potential evapotranspiration according to Eq. (4) ($E_{pot}$, H) and Eq. (5) ($E_{pot}$, D&A). The averages are for a 100−year period in an arid climate.





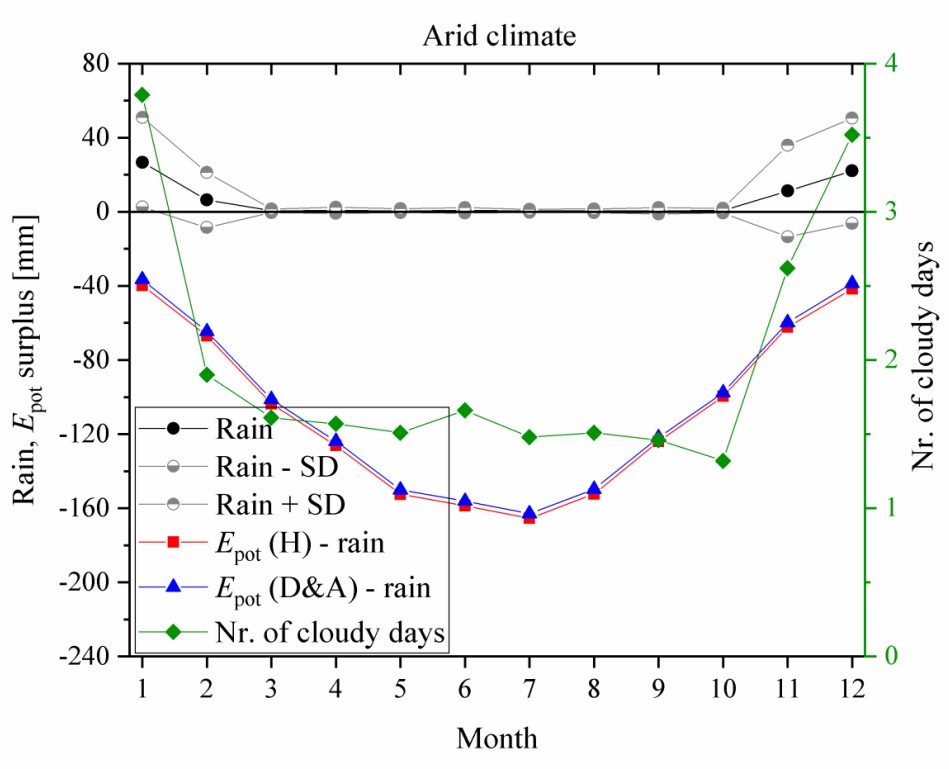


Figure 15. Average monthly values of the monthly rainfall, the number of cloudy days per month, and the monthly evapotranspiration deficit (−) according to Eq. (4) ($E_{pot}$, H − rain) and Eq. (5) ($E_{pot}$, D&A − rain). The averages are for a 100−year period in an arid climate.





Table 1. Parameters of the Truncated Bartlett−Lewis Gamma model (Pham et al., 2013) for rainfall
generation.

| Parameter | Units | Description |
|---|---|---|
| $\lambda$ | $d^{-1}$ | Parameter of the exponential distribution of intervals between storm starting times |
| $\alpha$ | | Shape parameter of the gamma−distribution of $\eta$, the parameter of the exponential distribution of the duration of rain cells in a storm |
| $\nu$ | d | Rate parameter of the gamma−distribution of $\eta$ |
| $\kappa$ | | $\kappa\eta$ is the parameter of the exponential distribution of intervals between rain cell starting times in a storm |
| $\varphi$ | | $\varphi\eta$ is the parameter of the exponential distribution of the duration of a storm |
| $p$ | | Shape parameter of the gamma−distribution of the rainfall rate of a rain cell |
| $\delta$ | $d\ mm^{-1}$ | Rate parameter of the gamma−distribution of the rainfall rate of a rain cell |
| $\varepsilon$ | $d^{-1}$ | Truncation parameter. Realizations of the gamma variate for $\eta < \varepsilon$ are rejected. |

If parameters were used to generate rainfall in mm h$^{-1}$, the original values of $\lambda$ and $\varepsilon$ need to be multiplied by 24, those of $\nu$ and $\delta$ need to be divided by 24.



Table 2. Potential evaporation rates calculated with the original (Eq. (4)) and modified Hargreaves
equations (Eq. (5)) for various combinations of daily mean temperature, daily temperature range, and
30−day total rainfall. The calculated values are relative to the value calculated with the original equation
for a mean temperature of 20°C and a daily range of 10°C.

| Average temperature (°C) | Temperature range (°C) | Original Hargreaves (Eq. (4)) | Modified Hargreaves (Eq. (5)) | |
|---|---|---|---|---|
| | | | 0 mm rain in 30 d | 60 mm rain in 30 d |
| 20 | 5 | 0.707 | 0.594 | 0.527 |
| | 10 | 1.000 | 1.007 | 0.950 |
| | 15 | 1.225 | 1.370 | 1.319 |
| 25 | 5 | 0.804 | 0.678 | 0.601 |
| | 10 | 1.138 | 1.148 | 1.083 |
| | 15 | 1.393 | 1.563 | 1.504 |
| 30 | 5 | 0.903 | 0.762 | 0.675 |
| | 10 | 1.277 | 1.291 | 1.218 |
| | 15 | 1.564 | 1.757 | 1.691 |





Table 3. Rainfall parameters for the moderate climate. Parameter values taken from Pham et al. (2013) for
Uccle (Belgium).

| Month | $\lambda\,(\mathrm{d}^{-1})$ | $\alpha$ | $\nu\,(\mathrm{d})$ | $\kappa$ | $\varphi$ | $p$ | $\delta\,(\mathrm{d\,mm^{-1}})$ | $\varepsilon\,(\mathrm{d}^{-1})$ |
|---|---|---|---|---|---|---|---|---|
| Jan | 0.768 | 3.000 | 0.032042 | 0.200 | 0.046 | 2.304 | 6.7958E−2 | 1.524E−14 |
| Feb | 0.672 | 3.000 | 0.031625 | 0.193 | 0.044 | 2.663 | 7.7708E−2 | 2.2224E−14 |
| Mar | 0.648 | 3.000 | 0.026750 | 0.223 | 0.044 | 1.463 | 4.1333E−2 | 1.3776E−14 |
| Apr | 0.648 | 3.000 | 0.019583 | 0.157 | 0.030 | 2.525 | 5.2542E−2 | 6.0000E−14 |
| May | 0.576 | 3.788 | 0.027417 | 0.167 | 0.035 | 0.696 | 2.9000E−2 | 4.0800E−12 |
| Jun | 0.552 | 5.292 | 0.045750 | 0.162 | 0.035 | 0.654 | 2.7250E−2 | 6.0480 |
| Jul | 0.576 | 5.893 | 0.044750 | 0.149 | 0.030 | 0.429 | 1.7875E−2 | 11.064 |
| Aug | 0.672 | 3.000 | 0.015083 | 0.217 | 0.046 | 0.716 | 2.9833E−2 | 2.9280E−13 |
| Sep | 0.600 | 3.000 | 0.017875 | 0.176 | 0.035 | 0.923 | 3.8458E−2 | 1.0224E−13 |
| Oct | 0.552 | 3.000 | 0.034333 | 0.166 | 0.038 | 1.523 | 6.3458E−2 | 3.8160E−14 |
| Nov | 0.696 | 3.000 | 0.034250 | 0.190 | 0.040 | 1.519 | 6.3292E−2 | 1.7184E−13 |
| Dec | 0.720 | 3.000 | 0.037333 | 0.180 | 0.043 | 1.936 | 8.0667E−2 | 2.2536E−14 |





Table 4. Rainfall data for the hypothetical arid climate.

| Period (start day – end day) | $\lambda$ (d$^{-1}$) | $\alpha$ | $\nu$ (d) | $\kappa$ | $\varphi$ | $p$ | $\delta$ (d mm$^{-1}$) | $\varepsilon$ (d$^{-1}$) |
|---|---|---|---|---|---|---|---|---|
| 304 − 31 | 0.120 | 3.0 | 0.750 | 1.0 | 0.125 | 6.0 | 2.0 | 1.0E−5 |
| 32 − 303 | 0.020 | 7.0 | 0.070 | 0.1 | 0.008 | 6.0 | 1.0 | 0.1 |




Table 5. Parameters for the temperature, cloudiness, and evapotranspiration. The parameters are explained in the main text.

| Parameter | Temperate climate | Arid climate | Equation(s) in which the parameter appears |
|---|---|---|---|
| $\bar{T}_a$ (°C) | 10.6 | 21.2 | 1a |
| $\sigma_T$ (°C) | 0.0 | 0.0 | 1a |
| $A_c$ (°C) | 6.0 | 9.0 | 1a |
| $A_o$ (°C) | 3.0 | 5.0 | 1a |
| $\sigma_a$ (°C) | 2.5 | 2.5 | 1a |
| $\psi$ (d) | −122 | −98 | 1a |
| $\phi$ | 0.6 | 0.6 | 1b |
| $\sigma_m$ (°C) | 2.0 | 1.5 | 1b |
| $\mu_f$ | 1.1949 | 1.488 | 2 |
| $\sigma_{f,c}$ | 0.2949 | 0.1764 | 2 |
| $\sigma_{f,o}$ | 0.1475 | 0.0882 | 2 |
| $P_l$ (mm) | 2.0 | 2.0 | 3 |
| $P_h$ (mm) | 20.0 | 20.0 | 3 |
| $f_1$ | 0.25 | 0.05 | 3 |
| $f_2$ | 0.95 | 0.50 | 3 |
| $f_3$ | 0.35 | 0.40 | 3 |
| Latitude (rad) | 0.8866 | 0.3977 | 6a,b |




Table 6. Rainfall statistics of the temperate and the arid climate.

| Period | Temperate climate | | Arid climate | |
|---|---|---|---|---|
| | Mean (mm) | Standard deviation (mm) | Mean (mm) | Standard deviation (mm) |
| Year | 480.5 | 69.1 | 71.3 | 50.8 |
| Jan | 64.6 | 26.1 | 26.7 | 24.2 |
| Feb | 52.3 | 22.0 | 6.4 | 14.9 |
| Mar | 60.4 | 28.2 | 0.58 | 0.98 |
| Apr | 54.4 | 22.3 | 0.75 | 1.6 |
| May | 24.0 | 10.5 | 0.60 | 1.0 |
| Jun | 25.1 | 10.0 | 0.73 | 1.4 |
| Jul | 22.3 | 10.1 | 0.48 | 0.86 |
| Aug | 21.0 | 10.4 | 0.56 | 0.99 |
| Sep | 21.6 | 9.7 | 0.54 | 1.7 |
| Oct | 37.8 | 16.7 | 0.65 | 1.3 |
| Nov | 47.0 | 19.9 | 11.2 | 24.7 |
| Dec | 50.0 | 18.4 | 22.1 | 28.4 |





Table 7. Annual mean values of various weather variables for the temperate and the arid climate.

| Variable | | Temperate climate | Arid climate |
|---|---|---|---|
| Temperature (°C) | mean | 10.74 | 21.27 |
| | daily minimum | 7.34 | 16.78 |
| | daily maximum | 14.2 | 25.76 |
| Nr of cloudy days | | 144 | 24.0 |
| Extraterrestrial radiation (equivalent mm water column) | | 3531 | 4948 |
| Potential evapotranspiration (mm) | Hargreaves (1994) | 629.9 | 1364 |
| | Droogers and Allen (2002) | 545.1 | 1334 |
| Evapotranspiration deficit (mm) | Hargreaves (1994) | 149.4 | 1293 |
| | Droogers and Allen (2002) | 64.5 | 1263 |
