# Peer review of "A simple weather generator for applications with limited data availability: TEmpotRain 1.0 for temperatures, extraterrestrial radiation, and potential evapotranspiration"

_Geoscientific Model Development, 2018_

## Referee Comment (RC1) · C. Kilsby (Referee) · 18 Jun 2018

This manuscript is a clear and well written report of the develement of a weather generator, with obvious utility for multiple applications as claimed. The model is well described, validated and demonstrated.

There is one major problem: it is essentially identical in concept, and very similar in practice, to the widely used UKCP09 weather generator developed in 2007, and made available on the internet since 2009. Since this work has been cited so widely (more

Interactive
comment

than 650 cites to the papers describing the concept and application) I am surprised not to see it referenced in the submitted paper, and comparison made to show any improvements.

The proposed weather generator uses the Bartlett-Lewis (BLRP) rainfall model, which is a functionally identical variant of the Neyman-Scott (NSRP) model used in the UKCP09 weather generator. I am also therefore surprised not to see the NSRP model (and its very widely used software realisation, RainSim - Burton et al 2008) referenced: this pre-dates the BLRP (first published by Kavvas and Delleur, 1975 and taken up by Rodriguez-Iturbe et al. in 1987). NSRP is more widely used and cited than BLRP: Scopus shows [NSRP 61 papers, cited by 1370] [BLRP 56 papers, cited by 1102] with BL/NS and (rainfall or precipitation) in the title/keywords/abstract.

My suggestion:

-the author should review the considerable body of work involving essentially identical methods following the UKCP09 weather generator development;

- if the approach can be shown to be substantially different and superior, then publication would be valuable - this is not the case in the present manuscript;

- otherwise, I think the work can only be a new contribution if a novel application (of essentially the same methodology) is presented.

Chris Kilsby, Newcastle University, June 2018

References

A daily weather generator for use in climate change studies, CG Kilsby, PD Jones, A Burton, AC Ford, HJ Fowler. . . - Environmental Modelling & Software, 2007 Cited by 416 (Google)

UK Climate Projections science report: Projections of future daily climate for the UK from the Weather Generator, PD Jones, CG Kilsby, C Harpham, V Glenis, A Burton -

[Figure]

University of Newcastle, UK, 2009 Cited by 243 (Google)

Jones, P. D., Harpham, C., Goodess, C. M., and Kilsby, C. G.: Perturbing a Weather Generator using change factors derived from Regional Climate Model simulations, Nonlin. Processes Geophys., 18, 503-511, https://doi.org/10.5194/npg-18-503-2011, 2011.

RainSim: A spatial–temporal stochastic rainfall modelling system, A Burton, CG Kilsby, HJ Fowler, PSP Cowpertwait, PE O'Connell Environmental Modelling & Software 23 (12), 1356-1369 Cited by 214 (Google)

Kavvas, M. L., and J. W. Delleur, The stochastic and chronologic structure of rainfall sequence: Application to Indiana, Tech. Rep. 57, Water Resour. Res. Cent., Purdue Univ., West Layfayette, Ind., 1975.

---

## Short Comment (SC1) · 22 Jun 2018

As explained in https://www.geoscientific-model-development.net/about/manuscript_types.html GMD is encouraging that authors upload the version of the program code of model described in manuscript (including relevant data sets) as a supplement or make the code and data available at a data repository preferable with an associated DOI (digital object identifier). Download from an institutional or personal web page is not encouraged but authors can still provide a reference to the website for newer versions or further information.

Lutz Gross GMD Executive Editor

---

## Author Comment (AC1) · 27 Jun 2018

Dear Executive Editor,

The supplement contains both codes and their executables, in compliance with the journal policy.

Your sincerely,

Gerrit de Rooij

---

## Referee Comment (RC2) · Anonymous Referee #2 · 29 Jun 2018

This is a concisely written paper covering a suite of models, about which the right amount of detail is generally provided. I have some general comments first.

The rainfall model that the authors use is a variant of a well-established model, the Modified or Random Parameter Bartlett-Lewis Rectangular Pulse model. The authors refer to the first paper in which this model was developed (Rodriguez-Iturbe & al., 1988, hereafter RCI88) which followed upon a seminal paper by the same authors the previous year (Rodriguez-Iturbe & al., 1987) which marked a step change in the approach to rainfall modelling. Previously, modellers had applied a variety of stochastic models

to the discrete time-series of hourly or daily rainfalls: these would however typically not perform well at other time-scales. The idea of modelling the unobserved continuoustime process in such a way that the statistics of the aggregation of this process to different time-scales could be derived analytically formed the basis the work in RCI88 in which the authors chose the Bartlett-Lewis point process as the basis of their new model, and this opened up a whole area of hydro-meteorological research into the use of Poisson-cluster processes (Bartlett-Lewis or Neyman-Scott) for rainfall modelling which is alive and well today.

On this topic, Reviewer 1 (R1) provides statistics of the number of papers and citations for the two well-known processes of that category, the Bartlett-Lewis and the Neyman-Scott. It is not clear what is being claimed with these figures, but there is a suggestion that comparisons of numbers of citations are reliable guides to the scientific quality of the papers. If that is the claim he is making, it is a self-defeating one: his review il-lustrates the (perfectly understandable) practice of reviewers drawing attention to their work which will then subsequently be cited by the authors in their revised paper. The number of citations therefore clearly depends on factors that are independent of scientific quality.

R1 argues that the authors should have included a reference to some paper using the Neyman-Scott process. This could indeed have been included by the authors, but in my view, it is not required here (and certainly not on grounds of historical precedence as I explained above). There are in fact many other approaches to rainfall modelling, aside from the very similar Neyman-Scott point process modelling approach: one could argue that these other approaches should have been included, had the authors carried out a proper review of approaches to rainfall modelling. Here, I'm thinking in particular of another approach that has a strong tradition extending as far back as the early development of Poisson-based approaches, but which differs in its fundamental philosophy. These are (multi-)fractal models (Schertzer and Lovejoy, 1987), typically random cascades in which the (multi-)scaling properties of the observed rainfall signal

GMDD
are modelled explicitly. Within that broad category there are also a range of options whose differences between them, and from the Point process approach, are more scientifically interesting than the minor difference between Neyman-Scott and Bartlett-Lewis process approaches. These are the issues of whether one should use bounded or unbounded cascades, macro-canonical or micro-canonical cascades (Menabde & Sivapalan, 2000), and about whether claims of universality for a certain type of multi-fractal approach are substantiated (Schertzer and Lovejoy, 1997). This approach to modelling is equally alive and well today (see Raut et al., 2018, and references therein – this is specifically a space-time model but the methodology is applicable to a purely temporal model). The question now is: should the authors have carried out such a proper review of stochastic rainfall model is but a component of a larger modelling strategy and that it is this combination of models is what is of importance and arguably novel here.

I say 'arguably' because it is important to flag the following: this approach is not novel in its outline at least, since the idea of modelling rainfall, then using the generated rainfall to model temperature and potential evaporation is at the heart of a well-known approach that is the UKCP09 weather generator (Kilsby et al., 2007). On this, I fully concur with R1's comment: a reference to this work is essential. The question then is whether the proposed suite of models under review is still sufficiently novel to warrant publication.

I have looked into the details of the UKCP09 weather generator to compare it with the method in the paper under review. I note the following:

(1) The rainfall generators are similar insofar as Neyman-Scott and Bartlett-Lewis processes are largely equivalent. But of course, there are different types of models under each heading whose mutual differences are often greater than those between these two approaches. It is therefore of interest to see whether the model chosen on the basis of a recent paper by Pham et al. (2013) has something to offer, e.g. with reGMDD
spect to extreme value reproduction which is often a problem for long return periods. This by itself, however, is not sufficiently novel as the Neyman-Scott approach use in Kilsby et al. (2007, hereafter K07) appears to perform well in terms of reproducing daily extremes (at least at the locations for which results are shown).

(2) As regards the temperature model, K07 uses AR(1) time-series models, after removing seasonality by normalising the temperatures (using means and standard deviations of half-monthly periods), for both the mean daily temperature and the temperature range. There are different models depending on whether we are considering two consecutive wet days, two consecutive dry days or wet-dry day transitions. All variables are assumed normal. In the paper under review, the seasonal trend is modelled parametrically, using a sinusoidal shape with parameters depending on whether the day is overcast or not. The stationary signal is then also modelled using an AR(1), with one such model for a clear day and one for an overcast day. The probability of an overcast or clear day is then dependent upon the rainfall amount (using a staircase function) The temperature range is modelled by a log-normal variable. So, here, there are similarities insofar as an AR(1) model is involved for the mean temperature, but the way this is used and the way the range is represented differ. The scheme used for the temperature range in particular seems to be important for the Potential Evapotranspiration (PET) through the extremes it generates (see lines 449-451), so the fact that it is log-normal here rather than normal as in K07 is likely to make a significant difference.

(3) Vapour pressure, sunshine duration and wind speed are then generated using linear regressions upon daily rainfall, mean temperature, temperature range and one another (this is a multi-variate regression) in K07. From this, the PET is derived using a version of the Penman formula. In the paper under review, a modified Hargreaves formula was preferred (and the authors explain why). The extra-terrestrial radiation required in that formula is then obtained using work published in 2013, so postdating K07. Here, there are no similarities in the methods.

On the basis of this analysis, I think that the detail of the combined model in the pa-
per under review is sufficiently distinct to be considered as a separate multi-model, although, as said above, it is based upon the same broad modelling idea of starting with the rainfall and conditioning the other variables upon it. It is therefore scientifically interesting to see how this different implementation of the same general approach performs (particularly given the significant differences in the PET scheme). Of course, and again I agree with R1 on this point, this calls for a comparison of the two implementations, but such a comparison cannot be required in this paper which contains enough material as it is.

Looking into the detail of the paper, I have the following comments:

LINE - COMMENT

118 The sentence is odd: 'other models (...) found that (...) models work well'. I think the first 'models' should be 'modellers'.

165 Formulae 6a and 6b contain products sin(xi) sin(xi) for cos⥹ and tan. These should be written as squares or is this a typo? Please check

210 It is not clear to the reader what 'adequate' might mean at this point, so a pointer to the further explanations in the paper would be helpful.

228 The authors specify that the model can be used to generate data for leap years. However, formulae 6a and 6c would seem to apply to non-leap years only. Please clarify.

230-250 These are issues of detail (e.g. how to generate random numbers from an exponential distribution) which could be moved to an appendix.

255-300 The procedure for selecting model parameters seems to involve a lot of choices, as the many 'set...' statements indicate. It seems that the idea here is to move away from a systematic calibration of the rainfall model, probably because of the difficulty of obtaining convergence of numerical optimisation schemes reported in the cited papers. But is it the case that the average non-rainfall specialist will have a clue

GMDD
as to how to set the required numbers to sensible values. The impression here is that too much is left to the expertise of the user. What about some guidelines as to what kind of values have been found reasonable by the authors? It may be the case that the information provided in the supplement and referred to in lines 352-355 addresses this. Please comment.

319-329 Here in the case of the temperature, useful guidance is provided for the user, so, referring back to my previous comment, the lack of it for the rainfall is all the more noticeable.

359-361 The reader will be somewhat unclear as to how rainfall parameters for the Sahara have been produced (i.e. the numbers in table 4, line 720), apparently without any data (?) This point is connected to the two previous ones. Please provide further explanations/guidelines here.

Finally, although the authors show many results of running the model suite. I am not clear overall, to what extent the model has been validated. There are comparisons between locations and seasons and comments about how the model produces what one might expect, but to what extent has the model been validated? This needs to be made clearer.

**REFERENCES**

Kilsby, C.G., Jones, P.D., Burton, A., Ford, A.C., Fowler, H.J., Harpham, C., James, P., Smith, A., Wilby, R.L. (2007) A daily weather generator for use in climate change studies, Environmental Modelling & Software, 22(12), 1705-1719

Menabde, M., Sivapalan, M. (2000) Modeling of rainfall time series and extremes using bounded random cascades and Levy-stable distributions, Water Resour. Res., 36(11), 3293-3300

Raut, B.A., Seed, A.W., Reeder, M.J., Jacob, C. (2018) A multiplicative caseade model for high-resolution space-time downscaling of rainfall, J. Geophys. Res. Atmos., 123,

GMDD
**2050-2067**

Rodriguez-Iturbe, I., Cox, D.R., Isham, V. (1987) Some models of rainfall based on stochastic point processes, Proc. Roy. Soc., A417, 283-298

Rodriguez-Iturbe, I., Cox, D.R., Isham, V. (1988) A point process model for rainfall: further developments, Proc. Roy. Soc., A410, 269-288

Schertzer, D., Lovejoy, S. (1987) Physical modeling and analysis of rain and clouds by anisotropic scaling multiplicative processes, J. Geophys. Res. Atmos., 92 (D8), 9693-9714

Schertzer, D., Lovejoy, S. (1997) Universal Multifractals Do Exist!: Comments on "A Statistical Analysis of Mesoscale Rainfall as a Random Cascade", J. Applied Meteor., 36, 1296-1303

+ references in the paper

---

## Author Comment (AC2) · 5 Jul 2018

Initial reply to Prof. Kilsby

I thank Prof. Kilsby for his review. In my reply, all references can be found in the discussion paper, unless stated otherwise.

Prof. Kilsby finds no problems in the presentation of the weather generator but points out that a similar weather generator (UKCP09) is not mentioned, and provides several references to that weather generator.

I carried out a web-search for weather generators and discussed the most pertinent ones in the Introduction. Why the search results did not include UKCP09 I do not know, but I thank Prof. Kilsby for bringing this to my attention. I concur with him that not discussing UKCP09 is an omission, and in the case the Editor allows me to revise the paper I will rectify this. The references included in the review serve as a valuable starting point for this. In the meantime I have also found a useful review paper comparing Bartlett-Lewis and Neyman-Scott processes.

Prof. Kilsby points to similarities in the structure of UKCP09 and TEmpotRain. I agree there are some similarities but submit that there are significant differences as well. Referee # 2 gives an astute description of these. In summary, the models for rainfall in UKCP09 and TEmpotRain are somewhat different, the models for temperature are quite different, and the models for potential evapotranspiration are completely different. The sequence in which the sub–models comprising the weather generators are executed is similar, but I believe that is a necessity when one chooses to use Poisson-process to generate rain storms (1st constraint). One has to start with the rainfall record and 'build' the rest of the weather around that. Because the daily temperature extremes feed into the evapotranspiration model (2nd constraint), the temperature record must be generated before the evapotranspiration record. These two constraints fix the order in which the model cascade has to be executed.

The Bartlett Lewis (BL) and the Neyman Scott (NS) approach to generate rain storms and rain cells within storms are too similar to justify a strong preference for either. For TEmpotRain, BL is slightly more advantageous because unlike NS, the BL process generates cell starting times that are in chronological order within a storm, thereby making the array sorting process that I implemented in the code in order to establish the continuous rainfall record more efficient. The citation analysis presented in Prof. Kilsby's review confirms the lack of a clear preference for either BL or NS, with a more or less even split in number of papers and citations. Referee 2 supports my choice of not reviewing theoretical work on rainfall generation, since GMD is not the outlet for

such work. Indeed, referee 2 provides a review in her/his publically available report of such quality that I wonder what I could add to it.

It should be noted that the model to generate potential evapotranspiration in TEmpotRain is much simpler than that in UKCP09. This was a deliberate choice in view of the applicability of the model in data-scarce areas. Droogers and Allen (2002) demonstrated that their modified Hargreaves equation outperformed the much more data-hungry Penman-Monteith equation if the underlying weather data had errors in them.

Apart from the differences in the submodels of TEmpotRain and UKCP09, there are marked differences in the operation of both weather generators. UKCP09 is trained for the UK, where impressive weather records are available. Application to other climate zones is possible but not straightforward. Kilsby et al. (2007; see reference in Prof. Kilsby's review report) indicate that the weather generator relies on regression relations between daily climatic variables and daily rainfall. They also point out that applying the model to other geographic locations requires suitable data to develop these regression relationships. Application to future climates would additionally require data from a regional climate model for the desired location. Kilsby et al. (2007) are aware of the heavy data requirement and mention poor data availability as a factor that prohibits the application of UKCP09. TEmpotRain is much more flexible in this respect. In response to referee 2 I intend to expand on the guidelines to select parameter values in case calibration data sets are not available to improve that flexibility even further.

The currently available web-based version of UKCP09 is an impressive feat. From the website and the manual I gained the impression that this version of the model has been exclusively designed for the UK, and I can easily imagine it has many users there. The web-based version generates multiple records of 100 years at most (30 years for hourly data), whereas TEmpotRain generates a single record of essentially arbitrary duration (I am currently generating 1000-year records). We are studying groundwater recharge in semi-arid regions and found that 100 years is not enough to capture rare catastrophic rainfall events or equally rare sequences of multiple wet years. We believe

that especially the latter phenomenon has the potential to provide recharge to deep aquifers, but we need to model the unsaturated flow process for several centuries to be able to capture this. For applications of this nature, the flexibility to generate records of arbitrary duration is very valuable. For this reason I included leap years in TEmpotRain. If one does not take into account February 29, summer and winter will have changed place after 730 years.

TEmpotRain was developed with the needs of soil and crop modelers in mind. For this group of potential users I expect it is more convenient to run a single longer simulation with a sequence of crops than it would to run a set of parallel 100-year simulations.

Given these operational differences between UKCP09 and TEmpotRain I believe both models have their own sets of users that do not overlap very much.

Gerrit H. de Rooij

---

## Author Comment (AC3) · 5 Jul 2018

Initial reply to Referee # 2

I thank referee 2 for the insightful review. The references in my reply below can all be found in the discussion paper.

Like Prof. Kilsby, this referee is satisfied with the style and structure of the paper. The referee also lists and weighs the contributions in the paper and concludes that sufficient material is there to warrant publication. To the contributions listed by the referee I would

like to add that the paper provides the first systematic comparison of the original and the modified Hargreaves equation (Hargreaves, 1994; Droogers and Allen, 2002), and provides a procedure to determine suitable parameter values in case data sets suitable for calibration are not available. The referee notes that more guidance for determining the parameters of the rainfall generator would be desirable. I have prepared a more comprehensive protocol for these parameters to comply with this recommendation. This includes for the first time a theoretical approach for estimating the cutoff parameter epsilon, which determines the minimum permitted value of parameter eta. In Pham et al. (2013), this parameter varies over 15 orders of magnitude throughout the year for a single location. In addition, the physical meaning of epsilon is less clear than that of the other parameters. Therefore, a theoretical basis for establishing the need to set the value of epsilon larger than zero and estimate its value is expected to be valuable for users.

Referee 2 discussed some points raised by Prof. Kilsby. According to referee 2 it is not necessary to review the various algorithms proposed in the literature to generate rainfall to support my choice for the Bartlett-Lewis model. I agree with this assessment, but nevertheless intend to at least discuss a review paper comparing the Bartlett-Lewis and Neyman-Scott algorithms. The referee provides an insightful overview of these and other algorithms, including useful references. In fact, I am uncertain how much a review by myself could add to what I consider to be an excellent discussion of the subject in the referee report. After all, this referee report is now part of the public record of this paper and readers can easily refer to the material therein (unfortunately without the possibility to cite the referee by name).

Referee 2 agrees with Prof. Kilsby that a reference to the UKCP09 weather generator is necessary. I agree with both reviewers and will take care of this if granted the opportunity to revise the paper.

Referee 2 adds a number of detailed comments. I checked them carefully. Several of them require minor modifications of the text to improve clarity. Others are related

to the protocol that I recommend to estimate the model parameters if insufficient data for calibration are available. As indicated above I expanded this protocol to address the concerns of the referee. The final remark asks about the validation of the model. A formal validation has not been performed. I believe all these comments can be addressed in a revision. If permitted to do so, I will provide a detailed reply explaining how I modified the text in response to these comments.

Gerrit H. de Rooij